# Action detection using a neural network elucidates the genetics of mouse grooming behavior

Brian Q Geuther, Asaf Peer, Hao He, Gautam Sabnis, Vivek M Philip, Vivek Kumar*

The Jackson Laboratory, Bar Harbor, United States

**Abstract** Automated detection of complex animal behaviors remains a challenging problem in neuroscience, particularly for behaviors that consist of disparate sequential motions. Grooming is a prototypical stereotyped behavior that is often used as an endophenotype in psychiatric genetics. Here, we used mouse grooming behavior as an example and developed a general purpose neural network architecture capable of dynamic action detection at human observer-level performance and operating across dozens of mouse strains with high visual diversity. We provide insights into the amount of human annotated training data that are needed to achieve such performance. We surveyed grooming behavior in the open field in 2457 mice across 62 strains, determined its heritable components, conducted GWAS to outline its genetic architecture, and performed PheWAS to link human psychiatric traits through shared underlying genetics. Our general machine learning solution that automatically classifies complex behaviors in large datasets will facilitate systematic studies of behavioral mechanisms.

*For correspondence:
Vivek.Kumar@jax.org

Competing interests: The authors declare that no competing interests exist.

## Introduction

Behavior, the primary output of the nervous system, is complex, hierarchical, dynamic, and high dimensional (*Gomez-Marin et al., 2014*). Precise approaches to dissect neuronal function require analysis of behavior at high temporal and spatial resolution. Achieving this is a time-consuming task and its automation remains a challenging problem in behavioral neuroscience. In the field of computer vision, modern neural network approaches have presented new solutions to visual tasks that perform just as well as humans (*Ching et al., 2018*; *Angermueller et al., 2016*). Application of these tools to biologically relevant problems could alleviate the costs of behavioral experiments and enhance reproducibility. Despite these enticing advantages, few aspects of behavioral biology research leverages neural network approaches. This lack of application is often attributed to the high cost of organizing and annotating the data sets, or to the stringent performance requirements. Thus, behavior recognition within dynamic environments is an open challenge in the machine learning community and translatability of proposed solutions to behavioral neuroscience remains unaddressed.

Behavioral action recognition falls under multiple types of computer vision problems, including action classification, event detection, and temporal action localization. Action classification, a task closely related to image captioning, trains a classifier to apply action labels to manually pre-trimmed video clips. This problem has already been largely solved, with the exceptional performance for networks competing in data sets such as Kinetics-400, Moments in Time, Youtube-8M, and many other available benchmark data sets (*Wu et al., 2017*). However, this classification does not determine when an action occurs within an untrimmed video. To address this shortcoming, two other tasks have been designed: event detection (ActivityNet 2019 Task 1) and temporal action localization (ActivityNet 2019 Task 2) (*Heilbron et al., 2015*). The objective of event detection is to identify

**eLife digest** Behavior is one of the ultimate and most complex outputs of the body's central nervous system, which controls movement, emotion and mood. It is also influenced by a person's genetics. Scientists studying the link between behavior and genetics often conduct experiments using animals, whose actions can be more easily characterized than humans. However, this involves recording hours of video footage, typically of mice or flies. Researchers must then add labels to this footage, identifying certain behaviors before further analysis.

This task of annotating video clips – similar to image captioning – is very time-consuming for investigators. But it could be automated by applying machine learning algorithms, trained with sufficient data. Some computer programs are already in use to detect patterns of behavior, however, there are some limitations. These programs could detect animal behavior (of flies and mice) in trimmed video clips, but not raw footage, and could not always accommodate different lighting conditions or experimental setups. Here, Geuther et al. set out to improve on these previous efforts to automate video annotation.

To do so, they used over 1,250 video clips annotated by experienced researchers to develop a general-purpose neural network for detecting mouse behaviors. After sufficient training, the computer model could detect mouse grooming behaviors in raw, untrimmed video clips just as well as human observers could. It also worked with mice of different coat colors, body shapes and sizes in open field animal tests.

Using the new computer model, Geuther et al. also studied the genetics underpinning behavior – far more thoroughly than previously possible – to explain why mice display different grooming behaviors. The algorithm analyzed 2,250 hours of video featuring over 60 kinds of mice and thousands of other animals. It found that mice bred in the laboratory groom less than mice recently collected from the wild do. Further analyses also identified genes linked to grooming traits in mice and found related genes in humans associated with behavioral disorders.

Automating video annotation using machine learning models could alleviate the costs of running lengthy behavioral experiments and enhance the reproducibility of study results. The latter is vital for translating behavioral research findings in mice to humans. This study has also provided insights into the amount of human-annotated training data needed to develop high-performing computer models, along with new understandings of how genetics shapes behavior.

when an event occurs, whereas the objective of temporal action detection is to identify where, when, and who is performing an action in untrimmed video input. The dominant approach for solving these issues has been extending region proposal methods from single images to video data. This involves proposing video tubelets (*Kalogeiton et al., 2017*; *Feichtenhofer et al., 2019*), a clip of video in both space and time for a single subject performing a single action.

In behavioral neuroscience, previous attempts to operate directly on visual data have utilized unsupervised behavioral clustering approaches (*Todd et al., 2017*). These include seminal work to convert visual data into frequency domains followed by clustering in *Drosophila* (*Berman et al., 2014*) and autoregressive Hidden Markov Model-based analysis of depth imaging data for mouse behavior (*Wiltschko et al., 2015*). Both approaches rely upon alignment of data from a top-down view and while they cluster similar video segments, interpretation of generated clusters is still dictated by the user. It is also unclear how these approaches will perform on sequences of disparate behaviors.

Supervised approaches in behavioral neuroscience have abstracted the subject into lower dimensions such as ellipse or key points, followed by feature generation, and classification (*Kabra et al., 2013*; *van den Boom et al., 2017*). While these approaches were a significant advance when they were introduced, they are inherently limited by the measurements available from the abstraction. For instance, standard measurements such as center of mass tracking, limit the types of behaviors that can be classified reliably. The field quickly recognized this issue and moved to integrate new measurements for the algorithms to classify behavior. These new features are highly specific to the organism and behavior that the researcher wishes to observe. In *Drosophila* studies, tracking of individual limbs and wings add new tracking modalities (*Robie et al., 2017*). For mice, modern systems

integrate floor vibration measurements and depth imaging techniques to enhance behavior detection (*Quinn et al., 2003*; *Hong et al., 2015*; *Wiltschko et al., 2015*). Vibration measurements set limits to both the environment and the number of animals, while depth imaging restricts the environment. While others have attempted to automate the annotation of mouse grooming using a machine learning classifier, available techniques are not robust for multiple animal coat colors, lighting conditions, and locations of the setup (*van den Boom et al., 2017*). Recent advances in computer vision also provide general purpose solutions for marker-less tracking in lab animals (*Mathis et al., 2018*; *Pereira et al., 2019*). These new techniques provide richer features to extend traditional machine learning techniques for behavioral classification. Human action detection leaderboards suggest that while the approach of pose estimation is powerful, it routinely underperforms compared to end-to-end solutions that utilize raw video input for action classification (*Feichtenhofer et al., 2019*; *Choutas et al., 2018*).

Here, we use neural networks to directly classify mouse grooming behavior from video. Grooming represents a form of stereotyped or patterned behavior of considerable biological importance consisting of a range of small to large actions. Grooming is an innate behavior conserved across animal species, including mammals (*Spruijt et al., 1992*; *Kalueff et al., 2010*). In rodents, a significant amount of waking behavior, between 20 and 50%, consists of grooming (*Van de Weerd et al., 2001*; *Spruijt et al., 1992*; *Bolles, 1960*). Grooming serves many adaptive functions such as coat and body care, stress reduction, de-arousal, social functions, thermoregulation, nociception, as well as other functions (*Spruijt et al., 1992*; *Kalueff et al., 2010*; *Fentress, 1988*). The neural circuitry that regulates grooming behavior has been studied, although much remains unknown. Importantly, grooming and other patterned behaviors are endophenotypes for many psychiatric illnesses. For instance, a high level of stereotyped behavior is seen in autism spectrum disorder (ASD), while in contrast, Parkinson's disease shows an inability to generate patterned behaviors (*Kalueff et al., 2010*). Therefore, the accurate and automated analysis of grooming behavior represents important value in behavioral neuroscience. We also reasoned that successful development of a neural network architecture for grooming behavior classification would be transferable to other behaviors by changing the training data.

We applied a general machine learning solution to mouse grooming and developed a classifier that performs at human level. This classifier performs across 62 inbred and F1 hybrid strains of mice consisting of visually diverse coat colors, body shapes, and sizes. We explored reasons why our network has an upper limit on performance that seems to be concordant with human annotations. Human level performance comes at a cost of a large amount of labeled training data. We identified environmental and genetic regulators of grooming behavior in the open field. Finally, we applied our grooming behavior solution to a genetically diverse mouse population and characterize the grooming pattern of the mouse in an open field. We used these data to carry out a genome wide association study (GWAS) and to identify the genetic architecture that regulates heritable variation in grooming and open-field behaviors in the laboratory mouse. Combined we propose a generalizable solution to complex action detection and apply it toward grooming behavior.

## Results

### Mouse grooming

Behavior varies widely on both time and space scales, from fine spatial movements such as whisking, blinking, or tremors to large spatial movements such as turning or walking, and temporally from milliseconds to minutes. We sought to develop a classifier that could observe and predict complex behaviors produced by the mouse. Grooming consists of syntaxes that are small or micro-motions (paw lick) to mid-size movements (unilateral and bilateral face wash) and large movements (flank licking) *Figure 1A*. There are also rare syntaxes such as genital and tail grooming. Grooming duration can vary from sub-seconds to minutes.

### Annotating grooming

Our approach to annotating grooming classified each frame in a video as the mouse being in one of two states: grooming or not grooming. We specified that a frame should be annotated as grooming when the mouse is performing any of the syntaxes of grooming, whether or not the mouse is

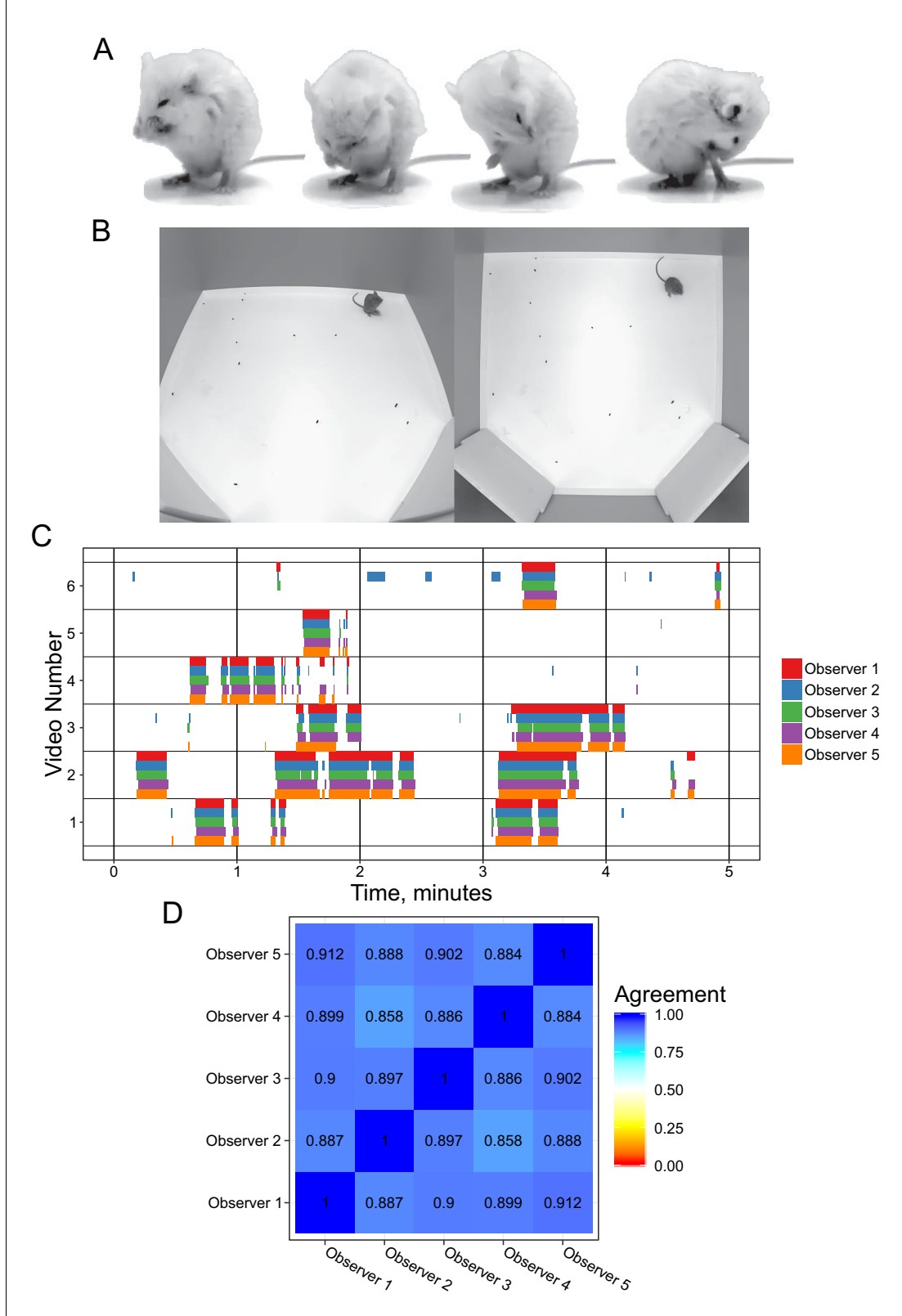

**Figure 1.** Annotation of mouse grooming behavior. (**A**) Mouse grooming contains a wide variety of postures. Paw licking, face-washing, flank linking, as well as other syntaxes all contribute to this visually diverse behavior. (**B**) We provided synchronized two-view data for observers to annotate. (**C**) Grooming ethograms for six videos by five different trained annotators. Overall, there is very high agreement between human annotators. (**D**) Quantification of the agreement overlap between individual annotators. Average agreement between all annotators is 89.13%.

*Figure 1 continued on next page*

*Figure 1 continued*

The online version of this article includes the following figure supplement(s) for figure 1:

**Figure supplement 1.** Additional details of annotator disagreements.

performing a stereotyped syntactic chain of grooming. This included a wide variety of postures and action durations which contribute to a diverse visual appearance. This also explicitly included individual paw licks as grooming, despite isolated paw licks not constituting a bout of grooming. Scratching was excluded from being classified as grooming.

We investigated the variability in manual grooming annotations by humans by tasking five trained annotators with labeling the same six 5 min videos (30 min total, *Figure 1*). To help human scorers, we provided these videos from a top-down and side view of the mouse (*Figure 1B*). These videos included C57BL/6J, BTBR and CAST/EiJ mouse strains. We gave each annotator the same instructions to label the behavior (see Methods). We observed a strong agreement (89.1% average) between annotators, which is in concordance with prior work annotating mouse grooming behavior (*Kyzar et al., 2011*). To examine disagreements between annotators, we classified them into three classes: missed bout, skipped break, and misalignment (*Figure 1—figure supplement 1*). Missed bout calls are made when a disagreement occurs in a not-grooming call. Similarly, skipped break calls are made when a disagreement occurs in a grooming call. Finally, misalignment is called when both annotators agree that grooming is either starting or ending but disagree on the exact frame in which this occurs. The most frequent type of error was misalignment, accounting for 50% of total duration of disagreement frames annotated and 75% of the disagreement calls (*Figure 1—figure supplement 1*).

Next, we constructed a large annotation data set to train a machine learning algorithm. While most machine learning contests seeking to solve tasks similar to ours have widely varied data set sizes, we leveraged network performance in these contests for design of our data set. Networks in these contests perform well when an individual class contains at least 10,000 annotated frames (*Girdhar et al., 2019*). As the number of annotations in a class exceeds 100,000, network performance for this task achieves mean average precision (mAP) scores above 0.7 (*Girdhar et al., 2019*; *Zhang et al., 2019*). With deep learning approaches, model performance benefits from additional annotations (*Sun et al., 2017*). To ensure success, we set out to annotate over 2 million frames with either grooming or not grooming. We aimed to balance this data set for grooming behavior by selecting video clips based on tracking heuristics, prioritizing segments with low velocity because a mouse cannot be grooming while walking. We also cropped the video frame to be centered on the mouse to reduce visual clutter using our tracker (*Geuther et al., 2019*). This cropping centered around the mouse follows the video tube approach, as seen in the current state of the art (*Feichtenhofer et al., 2019*). Based on this, we sampled 1253 short video clips from 157 videos. These video clips represent a diverse set of mice including 60 strains and a large range of body weights (*Figure 2—figure supplement 1A–B*). Using a pool of seven validated annotators, we obtained two annotations for each of the 1253 video clips totaling 2,637,363 frames with 94.3% agreement between annotations (*Figure 2A*).

## Proposed neural network solution

We trained a neural network classifier using our large annotated data set. Of the 1253 video clips, we held out 153 for validation. Using this split, we achieved similar distributions of frame-level classifications between training and validation sets (*Figure 2A*). Our machine learning approach takes video input data and produces an ethogram output for grooming behavior (*Figure 2B*). Functionally, our neural network model takes an input of 16 112 × 112 frames, applies multiple layers of 3D convolutions, 3D pooling, and fully connected layers to produce a prediction for only the last frame (*Figure 2C*). To predict a completed ethogram for a video, we slide the 16-frame window across the video.

We compared our neural network approach to a previously established machine learning approach for annotating lab animal behavior, JAABA (*Kabra et al., 2013*). Our neural network achieved 93.7% accuracy and 91.9% true positive rate (TPR) with a 5% false positive rate (FPR) (*Figure 3A,B*, pink line). In comparison, the JAABA trained classifier achieved a lower performance

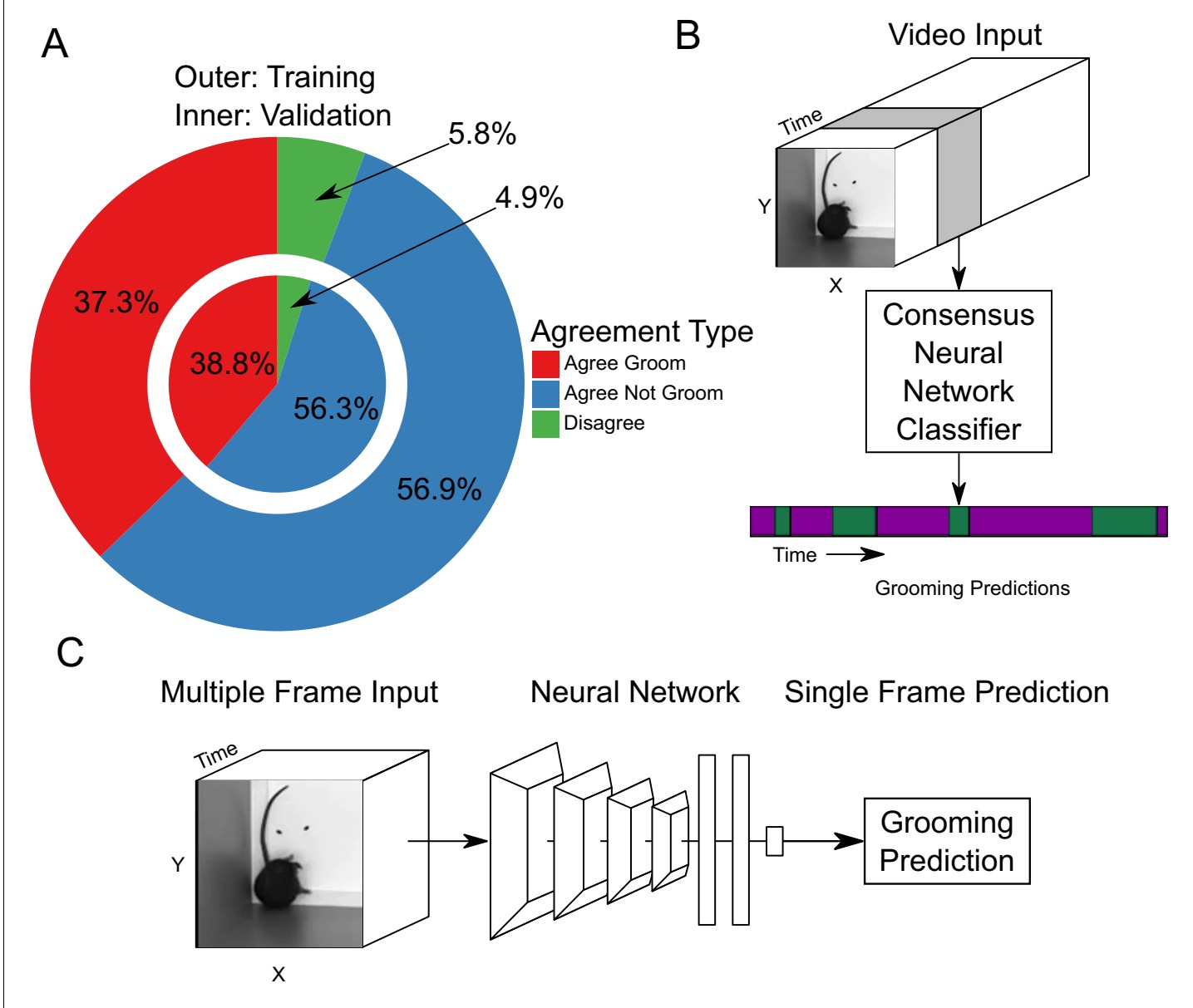

**Figure 2.** Neural network based action detection. (**A**) A total of 2,637,363 frames were annotated across 1253 video clips by two different human annotators to create this data set for training and analyzing our neural network. The outer ring represents the training data set agreement between human annotators while the inner ring represents the validation data set agreement between human annotators. (**B**) A visual description of the classification approach that we implemented. To analyze an entire video, we pass a sliding window of frames into a neural network. (**C**) Our network takes video input and produces a grooming prediction for a single frame.

The online version of this article includes the following figure supplement(s) for figure 2:

**Figure supplement 1.** Additional details about the annotated dataset.

of 84.9% accuracy and 64.2% TPR at a 5% FPR (*Figure 3A,B*). Due to memory limitations of JAABA, we could only train it using 20% of our training set. To test whether the training set size accounted for this poorer performance by JAABA, training our neural network using 20% of our training set still led to out-performance of JAABA (*Figure 3B*). When training the neural network using different sized training data sets, we observed improved validation performance with increasing data set size (*Figure 3—figure supplement 1A*). Scaling the training dataset size for JAABA showed that performance saturated when using 10% of our training data (*Figure 3—figure supplement 1B*). Using an interactive training protocol recommended by the authors of JAABA, we observed decreased

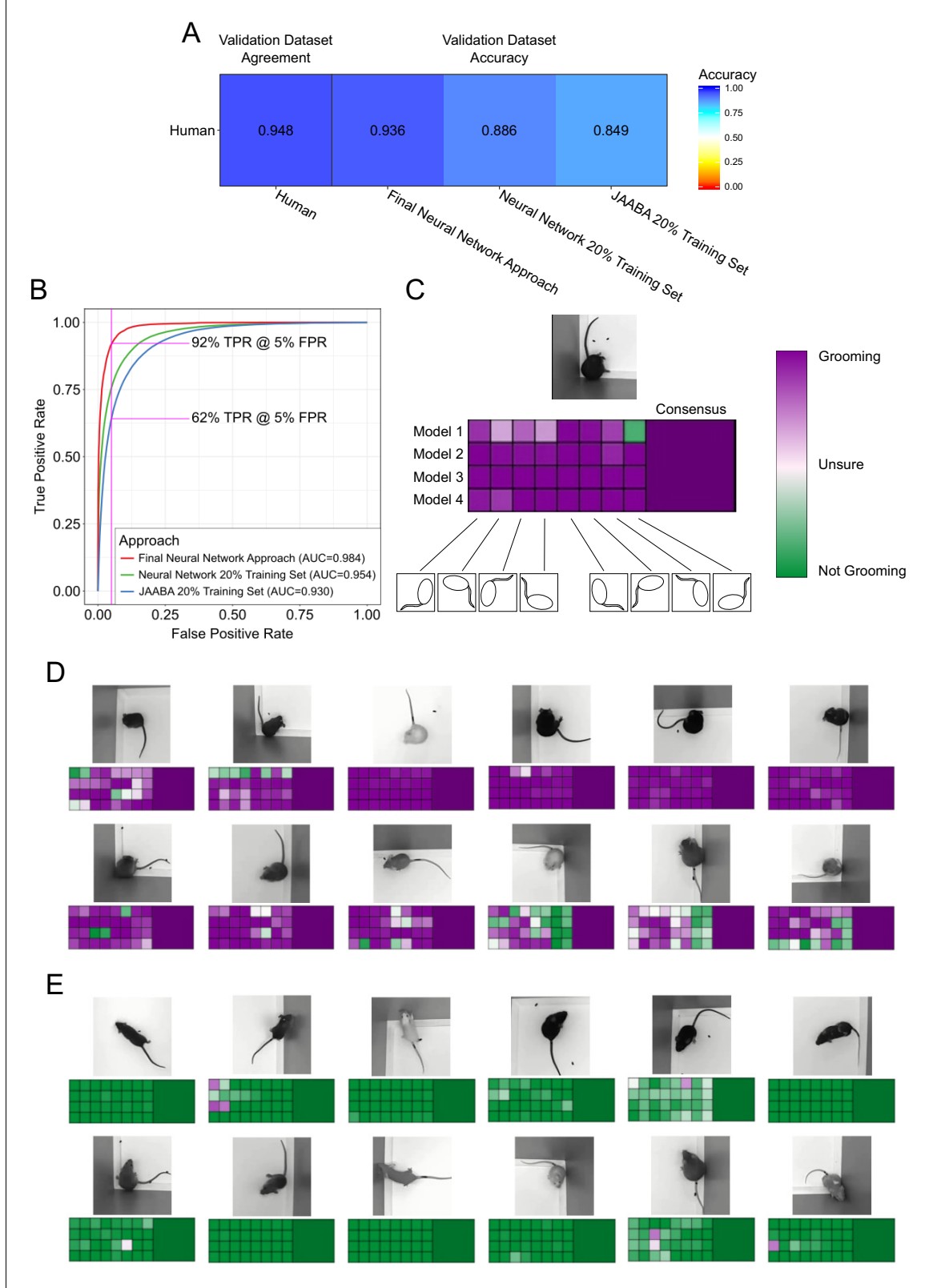

**Figure 3.** Validation of neural network model. (A) Agreement between the annotators while creating the data set compared to the accuracy of the algorithms predicting on this data set. We compared the machine learning models against only annotations where the annotators agree. (B) Receiver operating characteristic (ROC) curve for three machine learning techniques trained on the training set and applied to the validation set. Our final neural network model approach achieves the highest area under curve (AUC) value of 0.9843402. True positive at 5% False positive is indicated with pink line.

*Figure 3 continued on next page*

*Figure 3 continued*

(**C**) A visual description of our proposed consensus solution. We use a 32x consensus approach where we trained four separate models and give eight frame viewpoints to each. To combine these predictions, we averaged all 32 predictions. While one viewpoint from one model can be wrong, the mean prediction using this consensus improves accuracy. (**D–E**) Example frames where the model is correctly predicting grooming and not-grooming behavior. Also see *Figure 3—videos 1–9*.

The online version of this article includes the following video and figure supplement(s) for figure 3:

**Figure supplement 1.** Additional ROC curve subsets.
**Figure supplement 2.** Validation performance of algorithm split by video.
**Figure supplement 3.** Comparison of different consensus modalities and temporal smoothing.
**Figure 3—video 1.** Sample video of a black coat color mouse.
https://elifesciences.org/articles/63207#fig3video1
**Figure 3—video 2.** Sample video of an agouti coat color mouse.
https://elifesciences.org/articles/63207#fig3video2
**Figure 3—video 3.** Sample video of an agouti coat color mouse.
https://elifesciences.org/articles/63207#fig3video3
**Figure 3—video 4.** Sample video of an albino coat color mouse.
https://elifesciences.org/articles/63207#fig3video4
**Figure 3—video 5.** Sample video of an off-white coat color mouse.
https://elifesciences.org/articles/63207#fig3video5
**Figure 3—video 6.** Sample video of a gray coat color mouse.
https://elifesciences.org/articles/63207#fig3video6
**Figure 3—video 7.** Sample video of a chinchila coat color mouse.
https://elifesciences.org/articles/63207#fig3video7
**Figure 3—video 8.** Sample video of a nude coat color mouse.
https://elifesciences.org/articles/63207#fig3video8
**Figure 3—video 9.** Sample video of a kit mouse.
https://elifesciences.org/articles/63207#fig3video9

performance. This was likely due to the drastic size difference of the annotated data sets used in training (17,000 frames, or 0.7% of our annotated dataset). Interestingly, JAABA using 5% of our training dataset outperformed our neural network using 10% of our training dataset. This suggests that although JAABA may perform better using limited small datasets, both a neural network approach and a larger training dataset are necessary for generalizing on larger and more varied data.

Our neural network approach was as good as human annotators, given our previous observations in *Figure 1B–C* of 89% agreement. We inspected the receiver operating characteristic (ROC) curve performance on a per-video basis and found that performance was not uniform across all videos (*Figure 3—figure supplement 2*). The majority of the 153 validation videos were adequately annotated by both the neural network and JAABA. However, two videos performed poorly with both algorithms and seven videos showed drastic improvement using a neural network over the JAABA-trained classifier. Manual visual inspection of the two videos where both algorithms performed poorly suggests that they did not provide sufficient visual information to annotate grooming.

While developing our final neural network solution, we applied two forms of consensus modalities to improve single-model performance (*Figure 3C*). Each trained model makes slightly different predictions, due to randomness involved in training. This randomness appears in both network parameter initialization and the order of training batches. By training multiple models and merging the predictions, we achieved a slight improvement on validation performance. Additionally, we also modified the input image for different predictions. Rotating and reflecting the input image appears visually different for neural networks. We achieved 32 separate predictions for every frame by training four models and applying eight rotation and reflection transformations on the input. We merged these individual predictions by averaging the probability predictions. This consensus modality improved the ROC area under the curve (AUC) from 0.975 to 0.978. We attempted other approaches for merging the 32 predictions, including selecting the max value or applying a vote (median prediction). Averaging the prediction probabilities achieved the best performance (*Figure 3—figure supplement 3A*). Finally, we applied a temporal smoothing filter over 46 frames of

prediction. We identified 46 frames to be the optimal window for a rolling average (*Figure 3—figure supplement 3B*), which results in a final accuracy of 93.7% (ROC AUC of 0.984).

Our network can only make predictions using half a second worth of information. To ensure our validation performance is indicative of the wide diversity of mouse strains, we investigated the extremes of grooming bout predictions in our large strain survey data set which was not annotated by humans. While most of the long bout (>2 min) predictions were real, there were some false positives in which the mouse was resting in a grooming-like posture. To mitigate these rare false positives, we implemented a heuristic to adjust predictions. We experimentally identified that grooming motion typically causes ellipse-fit shape changes (W/L) to have a standard deviation greater than $2.5 \times 10^{-4}$. When a mouse is resting, the shape changes (W/L) standard deviation does not exceed $2 \times 10^{-5}$. Knowing that a mouse's posture in resting may be visually similar to a grooming posture, we assigned predictions in time segments where the standard deviation of shape change (W/L) over a 31 frame window was less than $5 \times 10^{-5}$ to a 'not grooming' prediction. Of all the frames in this difficult to annotate posture, 12% were classified as grooming. This suggests that this is not a failure case for our network, but rather a limitation of the network when only using half a second worth of information to make a prediction.

This approach handled varying mouse postures and physical appearance well, e.g. coat color and body weight. We observed good performance over a wide variety of postures and coat colors (*Figure 3C–D*, *Figure 3—videos 1–9*). Even nude mice, which have a drastically different appearance than other mice, achieved good performance. Visually, we observed instances where a small number of frame orientations and models make incorrect predictions. Despite this, the consensus classifier made the correct prediction.

## Definition of grooming behavioral metrics

We designed a variety of grooming behavioral metrics that describe both grooming quantity and grooming pattern. Following prior work (*Kalueff et al., 2010*), we defined a single grooming bout as continuous time spent grooming without interruption that exceeds 3 s. We allowed brief pauses (less than 10 s), but did not allow any locomotor activity for this merging of time segments spent grooming. Specifically, a pause occurred when motion of the mouse did not exceed twice its average body length. From this, we obtained a grooming ethogram for each mouse (*Figure 4A*). Using the ethogram, we summed the total duration of grooming calls in all grooming bouts to calculate the total duration of grooming.

Once we had the number of bouts and total duration, we calculated the average bout duration by dividing the two. For measurement purposes, we calculated the 5 min, 20 min, and 55 min summaries of these measurements. We included 5 and 20 min because these are typical open field assay durations.

Using 1 min binned data, we calculated a variety of grooming pattern metrics (*Figure 4B*). We fitted a linear slope to discover temporal patterning of grooming during the 55 min assay (GrTimeSlope55min). Positive slopes for total grooming duration infer that the individual mouse is increasing its time spent grooming the longer it remains in the open field test. Negative slopes for total grooming duration infer that the mouse spends more time grooming at the start of the open field test than at the end. This is typically due to the mouse choosing to spend more time doing another activity over grooming, such as sleeping. Positive slopes for number of bouts inferred that the mouse is initiating more grooming bouts the longer it remains in the open field test. Using 5 min binned data, we designed additional metrics to describe grooming pattern by selecting which 5 min bin a mouse spent the most time grooming (GrPeakMidBin) and the time duration spent grooming (GrPeakVal) in that minute. We also calculated a ratio between these values (GrPeakSlope). Finally, when we looked at strain-level averages of grooming, we identified how long a strain remains at its peak grooming (GrPeakLength).

We compared a variety of open field measurements including both grooming behavior and classical open field measurements (*Figure 4C*). We separated these 24 phenotypes into four groups. Grooming quantity describes how much an animal grooms, while grooming pattern metrics describe how an animal changes its grooming behavior over time. Open field anxiety measurements are traditional phenotypic measurements that have been validated to measure anxiety. Open field activity describes the general activity level of an animal.

## Sex and environment covariate analysis of grooming behavior

With this trained classifier, we sought to determine whether sex and environment affected grooming behavior in an open field, specifically grooming duration. We used data collected over 29 months for two strains, C57BL/6J and C57BL/6NJ to carry out this analysis. These two strains are substrains that were identical in 1951 and are two of the most widely used strains in mouse studies (*Bryant et al., 2018*). C57BL/6J is the mouse reference strain and C57BL/6NJ has been used by the International Mouse Phenotyping Consortium (IMPC) to generate a large amount of phenotypic data (*Brown and Moore, 2012*). We analyzed 775 C57BL/6J (317F, 458M) and 563 C57BL/6NJ (240F, 323M) mice tested over a wide variety of experimental conditions and ages. Across all these novel exposures in an open field, we quantified their grooming behavior for the first 30 min (*Figure 5*, 669 hr total data). We analyzed the data for effect of sex, season, time of day, age, room origin of the mice, light levels, tester, and white noise. To achieve this, we applied a stepwise linear model selection to model these covariates. Both forward and backward model selection results matched. After identifying significant covariates, we applied a second round of model selection that included sex interaction terms. The model selection identified sex, strain, room of origin, time of day, and season as significant. In contrast, age, weight, presence of white noise, and tester were not significant under our testing conditions. Additionally, the interaction between sex and both room of origin and season were identified as significant covariates.

| Covariate | p-value |
| --- | --- |
| Sex | $<2.2 \times 10^{-16}$ *** |
| Strain | 0.0267546 * |
| RoomOrigin | $5.357 \times 10^{-13}$ *** |
| Morning | 0.0001506 *** |
| Season | 0.0039826 ** |
| Sex by RoomOrigin | 0.0001568 *** |
| Sex by Season | 0.0235954 * |

We found an effect of strain (*Figure 5A*, $p = 0.0268$ C57BL/6J vs C57BL/6NJ) on grooming duration. Although the effect size was small, C57BL/6NJ groomed more than C57BL/6J. Additionally, we observed a sex difference (*Figure 5A*, $p<2.2 \times 10^{-16}$ males vs females). Males groomed more than females in both strains.

Since sex had a strong effect, we included interaction terms with other covariates in a second pass of our model selection. The model identified season as a significant covariate (*Figure 5B*, $p = 0.004$). Surprisingly, the model also identified an interaction between sex and season ($p = 0.024$). Female mice for both strains showed an increase in grooming during the summer and a decrease in the winter.

We carried out testing between 8AM and 4PM. To determine if the time of test affects grooming behavior, we split the data into two groups: morning (8am to noon) and afternoon (noon to 4pm). We observed a clear effect of time of day (*Figure 5C*, $p = 0.00015$). Mice tested in the morning groom more overall. We tested mice of different ages, ranging from 6 weeks to 26 weeks old. At the beginning of every test, we weighed the mice and found them to have a range of 16–42 g. We did not observe any significant effect of age (*Figure 5D*, $r = -0.065$, $p = 0.119$) or body weight ($r = 0.206$, $p = 0.289$) on grooming duration, although we did not test 'old' mice, generally considered to be more than 18 months old.

We compared the grooming levels of mice that were shipped from production rooms in a nearby building at our institution to our testing room with mice bred and raised in a room adjacent to the testing room (B2B). These production rooms contain a variety of possible confounding variables such as microbiome, noise, and technician-related stress. We specifically note that these room differences should not be due to genetic drift because of JAX's Genetic Stability Program , which periodically re-derives the strain from frozen embryos (*Taft et al., 2006*). Six production rooms supplied exclusively C57BL/6J (AX4, AX29, AX1, MP23, MP14, MP15), three rooms supplied exclusively C57BL/6NJ (MP13, MP16, AX5), and one room supplied both strains (AX8). All shipped mice were

housed in B2B for at least a week prior to testing. We observed a significant effect for room of origin (*Figure 5E*, $p = 5.357 \times 10^{-13}$). For instance, C57BL/6J males from AX4 and AX29 were low groomers compared to other rooms, including B2B. Shipped C57BL/6NJ from all rooms seemed to have low levels of grooming compared with B2B. We conclude that room of origin and shipping can both have effects on grooming behaviors.

We tested two light levels, 350–450 lux and 500–600 lux white light (5600K). We observed significant effects of light levels on grooming behavior (*Figure 5F*, $p = 0.04873$). Females from both strains groomed more in lower light, however males didn't seem to be affected. Despite this, our model did not include a light-sex interaction, suggesting that other covariates better account for the visual interaction with sex here.

The open field assays were carried out by one of two male testers, although the majority of tests were carried out by tester 2. Both testers carefully followed a testing protocol designed to minimize tester variation, which only involves weighing the mouse and placing the mouse into the arena. We observed no significant effect (*Figure 5G*, $p = 0.65718$) between testers.

Finally, white noise is often added to open field assays in order to create a uniform background noise levels and to mask noise created by the experimenter (*Gould, 2009*). Although the effects of white noise have not been extensively studied in mice, existing data indicate that higher levels of white noise increase ambulation (*Weyers et al., 1994*). We tested the effects of white noise (70 db) on grooming behavior of C57BL/6J and C57BL/6NJ mice and found no significant difference in duration spent grooming. Although there appears to be a stratification present for both C57BL/6J and C57BL/6NJ females, other cofactors better account for this.

Combined, these results indicate that environmental factors such as season, time of day, and room origin of the mice affect grooming behavior and may serve as environmental confounds in any grooming study. We also investigated age, body weight, light level, tester, and white noise and found these cofactors to not influence grooming behavior under our experimental conditions.

## Strain differences for grooming behavior

Next, we used the grooming classifier to carry out a survey of grooming behavior in the inbred mouse. We tested 43 classical laboratory and eight wild derived strains and 11 F1 hybrid mice from The Jackson Laboratory (JAX) mouse production facility. These were tested over a 31-month period and in most cases consisted of a single mouse shipment of mice from JAX production. Other than C57BL/6J and C57BL6/NJ, on average we tested eight males and eight females of an average age of 11 weeks for each strain. Each mouse was tested for 55 min in the open field as previously described (*Geuther et al., 2019*). This data set consisted of 2457 animals and 2252 hr of video. Video data were classified for grooming behavior as well as open field activity and anxiety metrics. Behavior metrics were extracted as described in *Figure 4*. In order to visualize the variance in phenotypes, we plotted each animal across all strains with corresponding strain mean and one standard deviation range and ethograms of select strains *Figure 6*. We distinguish between classical laboratory strains and wild derived inbred strains.

## Grooming amount and pattern in genetically diverse mice

We observed large continuous variance in total grooming time, average length of grooming bouts, and the number of grooming bouts in the 55 min open field assay (*Figure 6*). Total grooming time varied from 2 to 3 min in strains such as 129 × 1/SvJ and BALB/cByJ to 12 min in strains such as SJL/J and PWD/PhJ. Strains such as 129 × 1/SvJ and C57BR/cdJ had less than 10 bouts, whereas MA/MyJ had almost 40 bouts. The bout duration also varied from 5 s to approximately 50 s in BALB/cByJ and PWD/PhJ, respectively. In order to visualize relationships between phenotypes, we created strain mean and 1SD range correlation plots (*Figure 7*). There was a positive correlation between the total grooming time and the number of bouts as well as the total grooming time and average bout duration. Overall, strains with high total grooming time had increased number of bouts as well as longer duration of bouts. However, there did not seem to be a relationship between the number of bouts and the average bout duration, implying that the bout lengths stay constant regardless of how many may occur (*Figure 7*). In general, C57BL/6J and C57BL6/NJ fall roughly in the middle for classical inbred strains.

We investigated the pattern of grooming over time by constructing a rate of change in 5 min bins for each strain (*Figure 8*). There appeared to be visual structures in the data, so we used k-means to identify clusters. We identified three clusters of grooming patterns based on total grooming level and relative changes in 5 min binned grooming data (*Figure 1—figure supplement 1*). Type 1 consists of 13 strains with an 'inverted U' grooming pattern. These strains escalate grooming quickly once in the open field, reach a peak, and then start to decrease the amount of grooming, usually leading to a negative overall grooming slope. Often, we find animals from these strains are sleeping by the end of the 55 min open field assay. These strains include both high groomers such as CZE-CHII/EiJ, MOLF/EiJ, and low groomers such as 129 × 1/SvJ and I/LnJ. Type 2 consists of 12 strains that are high grooming strains and do not reduce grooming by the end of the assay. They reach peak grooming early (e.g. PWD/PhJ, SJL/J and BTBR) or late (e.g. DBA/2J, CBA/J) and then remain at or near this peak level for the remainder of the assay. The defining feature of this group is that a high level of grooming is maintained throughout the assay. Type 3 consists of most of the strains (30) and shows steady increase in grooming until the end of the assay. Overall, the strains in this group are medium-to-low groomers with a constant low positive or flat slope. We conclude that under our experimental conditions there are at least three broad, albeit continuous, classes of observable grooming patterns in the mouse.

## Wild derived vs. classical strain grooming patterns

We compared grooming patterns between classical and wild derived laboratory strains. Classical laboratory strains are derived from limited genetic stock originating from Japanese and European mouse fanciers (*Keeler, 1931*; *Morse, 1978*; *Silver, 1995*). Classical laboratory inbred mouse lines represent the genome of *Mus musculus domesticus*(*M.m domesticus*) 95% and *Mus musculus musculus* 5% (*Yang et al., 2011*). New wild derived inbred strains were established specifically to overcome the limited genetic diversity of the classical inbred lines (*Guénet and Bonhomme, 2003*; *Koide et al., 2011*). We observed that most wild derived strains groom for significantly higher duration and have longer average bout length than the classical inbred strains. Five of the highest 16 grooming strains are wild derived (PWD/PhJ, WSB/EiJ, CZECHII/EiJ, MSM/MsJ, MOLF/EiJ in *Figure 6A*). The wild derived strains also had significantly longer bouts of grooming, with 6 of 16 longest average grooming bout strains from this group. Both the total grooming time and average bout length were significantly different between classical and wild-derived strains (*Figure 1—figure supplement 1*). These high grooming strains represent *M.m. domesticus* and *M.M. musculus* subspecies, which are the precursors to classical laboratory strains (*Yang et al., 2011*). These wild derived strains also represent much more of the natural genetic diversity of the mouse populations than the larger number of classical strains we tested. This leads us to conclude that the high levels of grooming seen in the wild derived strains better represent the normal levels of grooming behavior in mice. This implies that low grooming behavior may have been selected for in classical laboratory strains, at least as observed in our experimental conditions.

## BTBR grooming pattern

We also closely examined the grooming patterns of the BTBR strain, which has been proposed as a model with certain features of autism spectrum disorder (ASD). ASD is a complex neurodevelopmental disorder leading to repetitive behaviors and deficits in communication and social interaction (*Association, 2013*). Compared to C57BL/6J mice, BTBR have been shown to have high levels of repetitive behavior, low sociability, unusual vocalization, and behavioral inflexibility (*McFarlane et al., 2008*; *Silverman et al., 2010*; *Moy et al., 2007*; *Scattoni et al., 2008*). Repetitive behavior is often assessed by self grooming behavior, and drugs with efficacy in alleviating symptoms of repetitive behavior in ASD also reduce grooming in BTBR mice without affecting overall activity levels, which provides some level of construct validity (*Silverman et al., 2012*; *Amodeo et al., 2017*).

We found that total grooming time in BTBR is high compared with C57BL/6J but is not exceptionally high compared to all strains (*Figure 6*), or even among classical inbred strains. C57BL/6J mice groomed approximately 5 min over a 55 min open field session, whereas BTBR groomed approximately 12 min (*Figure 6A*). Several classical inbred strains had similar levels of high grooming, including SJL/J, DBA/1J, and CBA/CaJ. The grooming pattern of BTBR belongs to Type 2 which

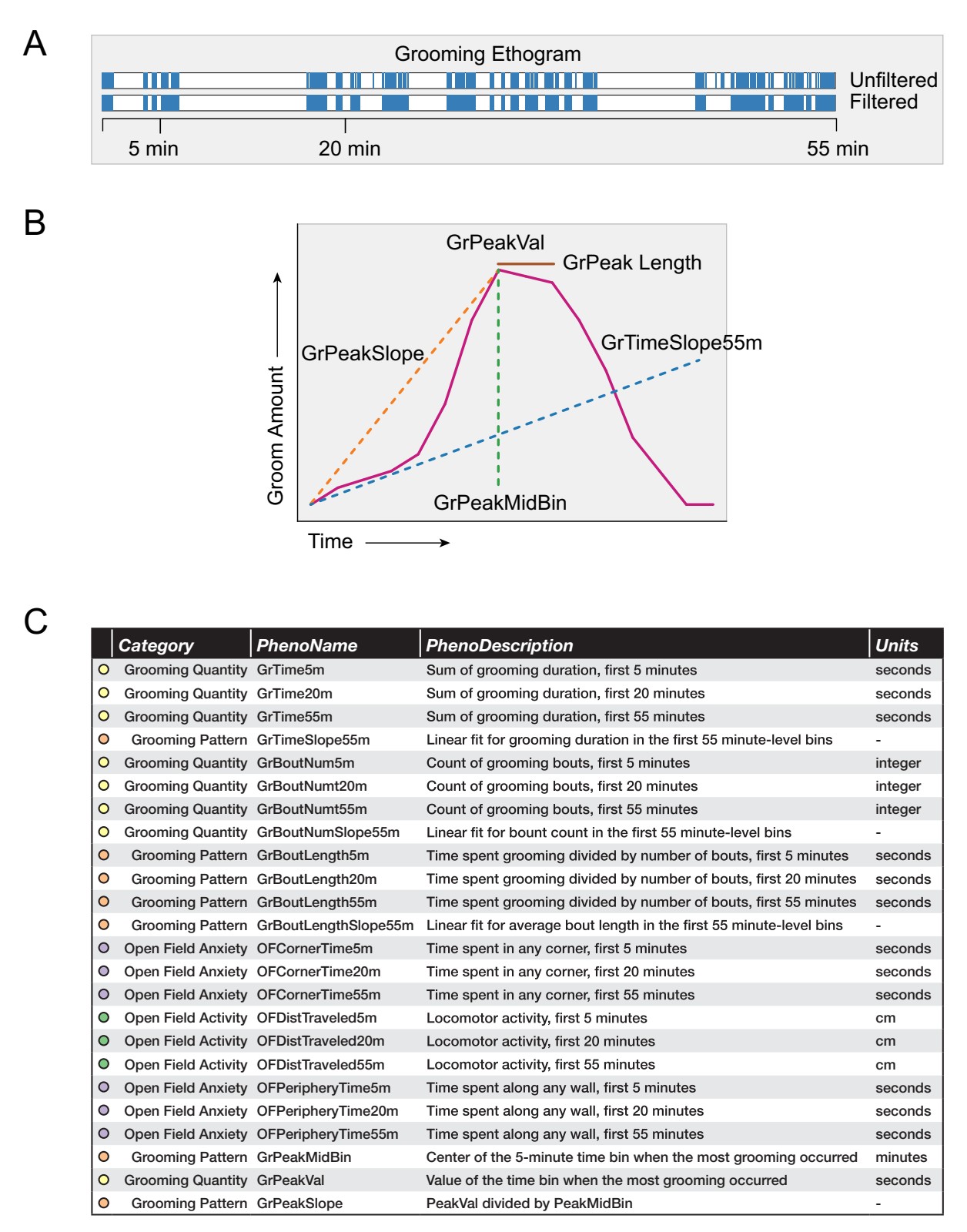

**Figure 4.** Grooming and open field behavioral metrics. (**A**) Example grooming ethogram for a single animal. Time is on the x-axis and a blue colored bar signifies that the animal was performing grooming behavior during that time. We calculated summaries at 5, 20, and 55 min ranges. (**B**) A visual description of how we define our grooming pattern phenotypes. (**C**) A table summarizing the 24 behavioral metrics we analyzed. We grouped the phenotypes into four groups, including grooming quantity, grooming pattern, open field anxiety, and open field activity.

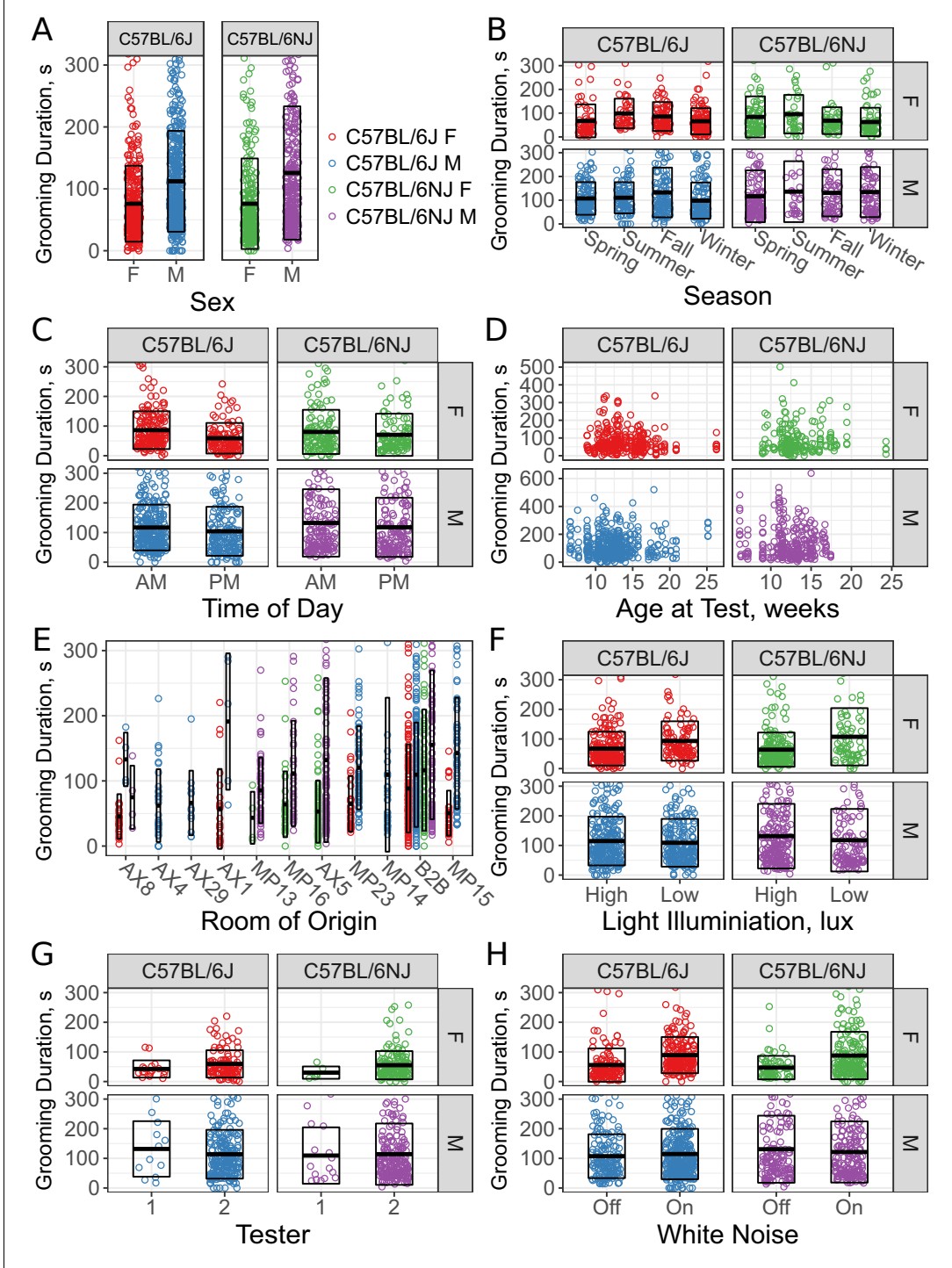

**Figure 5.** Sex and environmental covariate analysis of grooming behavior (total time grooming) in the open field for C57BL/6J and C57BL/6NJ strains. Effect of sex (A), season (B), time of day (C), age (D), room of origin (E), light level (F), tester (G), and white noise (H).

contains 11 other strains (*Figure 8*). One distinguishing factor of BTBR mice is that they have longer average bouts of grooming from an early point in the open field (*Figure 6—figure supplement 1*). However, again they were not exceptionally high in average bout length measure (*Figure 6—figure supplement 1*). Strains such as SJL/J,PWD/PhJ, MOLF/EiJ, NZB/BlNJ had similar long bouts from an early point. We conclude that BTBR display high levels of grooming with long grooming bouts,

however, this behavior is similar to several wild derived and classical laboratory inbred strains and is not exceptional. Since we did not measure social interaction and other salient features of ASD, we do not argue against BTBR as an ASD model. In addition to BTBR, perhaps other strains from the type two group could should be explored as ASD models.

## Grooming mouse GWAS

Next we wished to understand the underlying genetic architecture of complex mouse grooming behavior and open field behaviors, and to relate these to human traits. We used the data from the 51 inbred strains and 11 F1 hybrid strains to carry out a genome wide association study (GWAS). We did not include the eight wild derived strains because they are highly divergent and can skew mouse GWAS analysis. We analyzed the 24 phenotypes categorized into four categories – open field activity, anxiety, grooming pattern, and quantity (*Figure 4*). We used a linear mixed model (LMM) implemented in Genome-wide Efficient Mixed Model Association (GEMMA) for this analysis in order to control for spurious association due to population structure (*Zhou and Stephens, 2012*). We first calculated heritability of each phenotype by determining the proportion of variance in phenotypes explained by the typed genotypes (PVE) *Figure 9A*. Heritability ranged from 6% to 68%, and 22/24 traits have heritability estimates greater than 20%, a reasonable estimate for behavioral traits in mice and humans (*Valdar et al., 2006*; *Bouchard, 2004*), and makes them amenable for GWAS analysis (*Figure 9A*).

We analyzed each phenotype using GEMMA, and considered the resulting Wald test p-value. In order to correct for the multiple SNPs we tested (222,966), and to account for the correlations between SNPs genotypes, we obtained an empirical threshold for the p-values by shuffling the values of one normally distributed phenotype (OFDistTraveled20m) and taking the minimal p-value of each permutation. This process resulted in a p-value threshold of $1.4 \times 10^{-5}$ that reflects a corrected p-value of 0.05 (*Belmonte and Yurgelun-Todd, 2001*). We defined quantitative trait loci (QTL) in the following manner: adjacent SNPs that have correlated genotypes ($r^2 > = 0.2$) were clustered together in a greedy way, starting with the SNP with the lowest p-value in the genome, assigning it a locus, adding all correlating SNPs and then moving forward to the next SNP with the lowest p-value until all the significant SNPs are assigned to QTL. The genetic architecture of inbred mouse strains dictates large linkage disequilibrium (LD) blocks (*Figure 9B*), resulting in QTL that span millions of base-pairs and contain multiple genes (*Supplementary file 2*).

GWAS analysis of each phenotype resulted in 2 to 22 QTL (*Figure 9—figure supplement 1*, *Figure 9—figure supplement 2*, *Figure 9—figure supplement 3*, *Figure 9—figure supplement 4*, *Figure 9—figure supplement 5*, *Figure 9—figure supplement 6*). Overall, the open field activity had 15 QTL, anxiety 10, grooming pattern 76 and grooming quantity 51 QTL, leading to 130 QTL combined over all the tested phenotypes (*Figure 9C*). We observed pleiotropy with the same loci significantly associated with multiple phenotypes. Pleiotropy is expected since many of our phenotypes are correlated and individual traits may be regulated by similar genetic loci. For instance, we expected pleiotropy for grooming time in 55 and 20 min (GrTime55 and GrTime20) since these are correlated traits. We also expected that some loci regulating open field activity phenotypes may regulate grooming. In order to better understand the pleiotropic structure of our GWAS results, we generated a heat map of significant QTL across all phenotypes. We then clustered these, to find sets of QTL that regulate groups of phenotypes (*Figure 9D*). The phenotypes were clustered into five subgroups consisting of grooming pattern (I), open field activity (II), open field anxiety (III), grooming length (IV), and grooming number and amount (V) (*Figure 9D* top x-axis). We found seven clusters of QTL that regulate combinations of these phenotypes (*Figure 9D* y-axis). For instance, clusters A and G are composed of pleiotropic QTL that regulate grooming length (IV) and grooming time but QTL in cluster G also regulate bout number and amount (V). QTL cluster D regulates open field activity and anxiety phenotypes. Cluster E contains QTL that regulate grooming and open field activity and anxiety phenotypes, but most of the SNPs only have significant p-values for either open field phenotypes or grooming phenotypes but not both, indicating that independent genetic loci are largely responsible for these phenotypes. We colored the associated SNPs in the Manhattan plot (*Figure 9C*) with colors to mark one of these seven QTL clusters (*Figure 9D*). These clusters ranged from 13 to 35 QTL, with the smallest being cluster F which is mostly pleiotropic for grooming number, and the largest cluster, cluster G, is pleiotropic for most of the grooming related phenotypes.

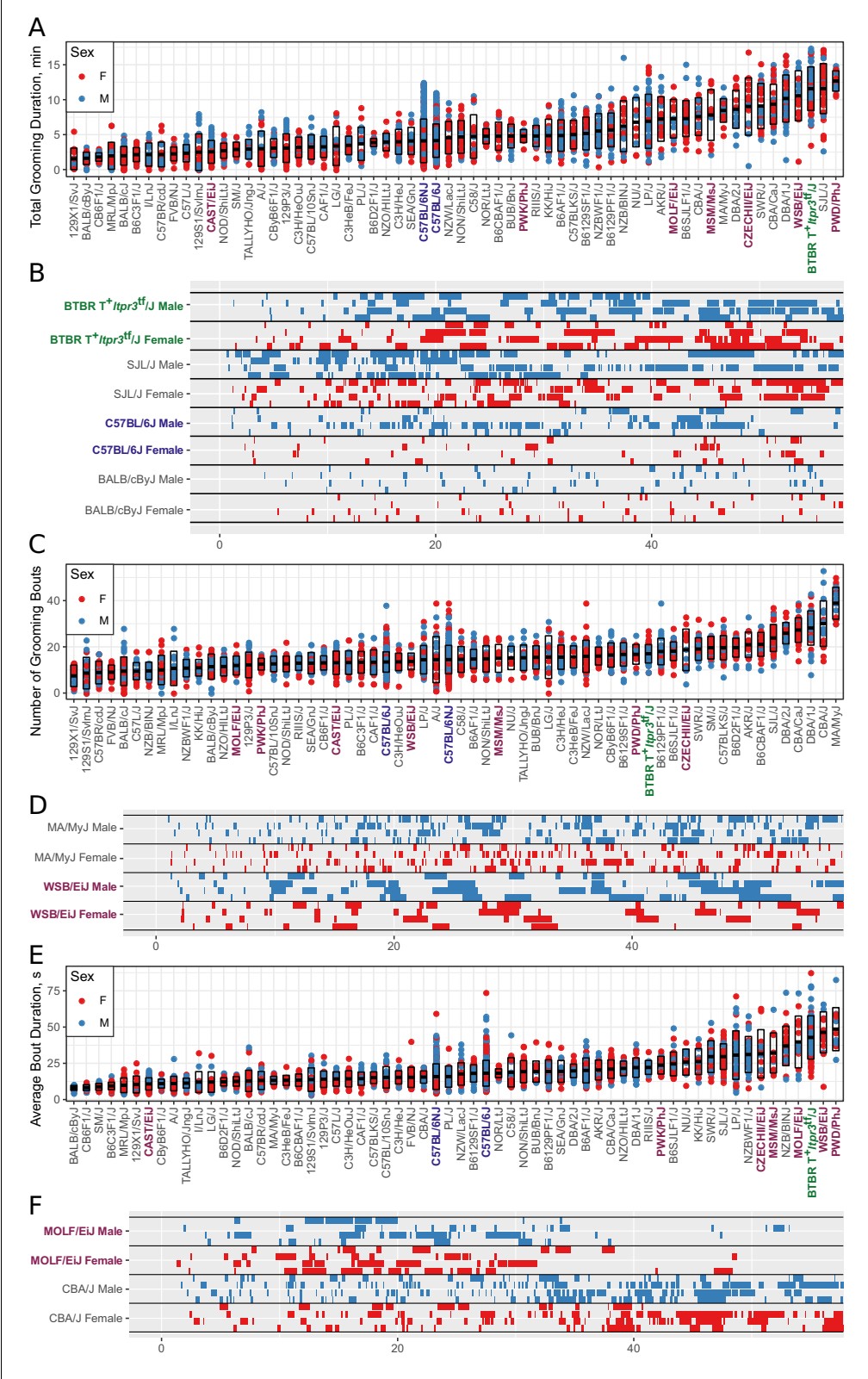

**Figure 6.** Strain survey of grooming phenotypes with representative ethograms (2457 animals and 2252 hr of data, each dot represents 55 min of a single animal). (**A**) Strain survey results for total grooming time. Strains present a smooth gradient of time spent grooming, with wild derived strains (purple) showing enrichment on the high end. (**B**) Representative ethograms showing strains with high and low total grooming time. (**C**) Strain survey results for the number of grooming bouts. (**D**) Comparative ethograms for two strains with different number of bouts, but similar total time spent

*Figure 6 continued on next page*

*Figure 6 continued*

grooming. (E) Strain survey results for average grooming bout duration. (F) Comparative ethograms for two strains with different average bout length, but similar total time spent grooming.

The online version of this article includes the following figure supplement(s) for figure 6:

**Figure supplement 1.** Grooming in wild-derived vs. classical inbred lines.

These highly pleiotropic genes included several genes known to regulate mammalian grooming, striatal function, neuronal development, and even language. Mammalian Phenotype Ontology Enrichment showed 'nervous system development' as the most significant module with 178 genes ($p = 7.5 \times 10^{-4}$) followed by preweaning lethality ($p = 3.5 \times 10^{-3}$, 189 genes) and abnormal embryo development ($p = 5.5 \times 10^{-3}$, 62 genes) (**Supplementary file 2**). We carried out pathway analysis using KEGG and Reactome databases (**Ogris et al., 2016**). This analysis showed 14 disease pathways that are enriched including Parkinson's ($9.68 \times 10^{-9}$), Huntington's ($1.07 \times 10{-6}$), non-alcoholic fatty liver disease ($9.31 \times 10^{-6}$), and Alzheimer's ($1.15 \times 10^{-5}$) as the most significantly enriched. Enriched pathways included oxidative phosphorylation ($6.42 \times 10^{-8}$), ribosome (0.00000102), RNA transport (0.00000315), and ribosome biogenesis (0.00000465). Reactome enriched pathways included mitochondrial translation termination and elongation ($2.50 \times 10^{-19}$ and $5.89 \times 10^{-19}$, respectively), and ubiquitin-specific processing proteases ($1.86 \times 10^{-8}$). The most pleiotropic gene was Sox5 which associated with 11 grooming and open field phenotypes. *Sox5* has been extensively linked to neuronal differentiation, patterning, and stem cell maintenance (**Lefebvre, 2010**). Its dysregulation in humans has been implicated in Lamb-Shaffer syndrome and ASD, both neurodevelopmental disorder (**Kwan, 2013**; **Zawerton et al., 2020**). A total of 102 genes were associated with 10 phenotypes, and 105 genes were associated with nine phenotypes. We limited our analysis to genes with at least six significantly associated phenotypes, resulting in 860 genes. Other genes include *FoxP1*, which has been linked to striatal function and regulation of language (**Bowers and Konopka, 2012**). *Ctnnb1*, a regulator of Wnt signaling, and *Grin2b*, a regulator of glutamate signaling. Combined, this analysis indicated genes known to regulate nervous system function and development, and genes known to regulate neurodegenerative diseases as regulators of grooming and open field behaviors. The GWAS also begins to define the genetic architecture of grooming and open field behaviors in mice.

## PheWAS

Finally, we wanted to link genes that are associated with open field and grooming phenotypes in the mouse with human phenotypes. We hypothesized that common underlying genetic and neuronal architectures exist between mouse and human, however, they can give rise to disparate phenotypes in each organism. For example, the misregulation of a pathway in the mouse may lead to over-grooming phenotype but in humans the same pathway perturbation may manifest itself as neuroticism or obsessive compulsive disorder. These relationships between phenotypes can be revealed through identification of common underlying genetic architectures. In order to link human and mouse phenotypes, we carried out PheWAS analysis with the 860 mouse grooming and open field genes with at least 6 degrees of pleiotropy identified in the mouse GWAS. We identified 509 human orthologs out of 860 mouse genes and downloaded PheWAS summary statistics from gwasATLAS. The gwasATLAS (Release 2: v20190117) contains 4756 GWAS from 473 unique studies across 3302 unique traits which are classified into 28 general domains and 192 subchapters obtained from either ICD10 or ICF10 (**Watanabe et al., 2019**). For each human ortholog, we focused on the association in the gwasATLAS Psychiatric domain with gene-level p value $\leq$ 0.001. We then turned the relationships between genes and psychiatric traits into a weighted bipartite network, where the weight of an edge is represented by the association strength (-log10(p value)) between a gene and a trait. We identified eight gene-phenotype modules within this weighted bipartite networks (**Figure 10**). These modules contained between 15 and 32 individual phenotypes and between 41 and 103 genes. At the subchapter level, modules were enriched for temperament and personality phenotypes, mental and behavioral disorders (schizophrenia, bipolar, dementia), addiction (alcohol, tobacco, cannabinoid) obsessive-compulsive disorder, anxiety, and sleep.

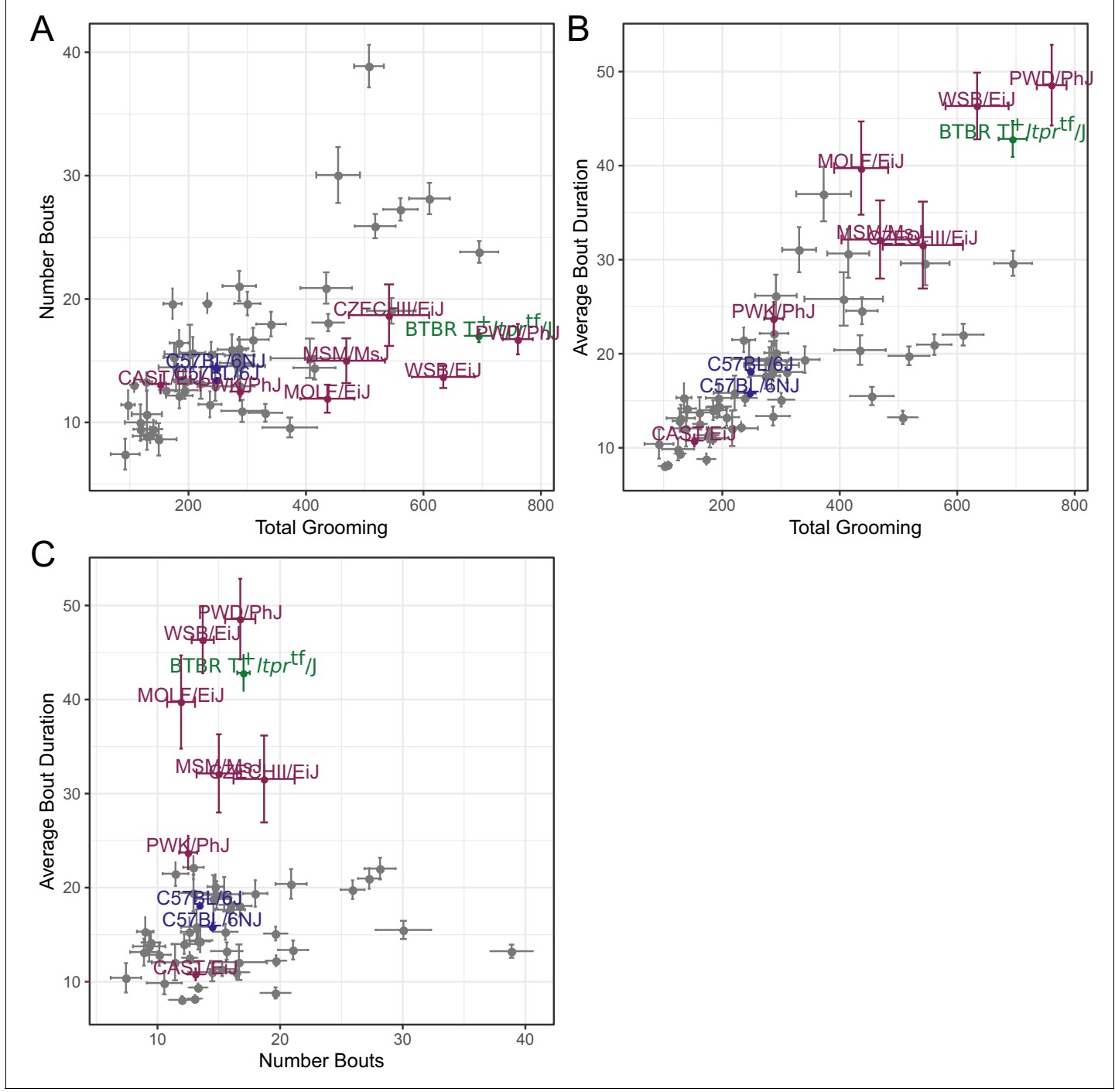

**Figure 7.** Relatedness of grooming phenotypes. Points indicate strain-level means. Lines indicate 1SD. (**A**) Strain survey comparing total grooming time and number of bouts. Wild derived strains and BTBR show enrichment for having high grooming but low bout numbers. (**B**) Strain survey comparing total grooming time and average bout duration. Strains that groom more also tend to have a longer average bout length. (**C**) Strain survey comparing number of bouts to average bout duration.

Surprisingly, we found orthologs that show high levels of pleiotropy in mouse GWAS and the resulting human PheWAS. FOXP1 is the most pleiotropic gene with 35 human phenotypic associations, while SOX5 is second with 33 associations. In order to prioritize candidate modules for further research, first, we produced a ranked list of modules by calculating modularity score, which measures the strength of division of a network into modules. The high-ranking modules represent most

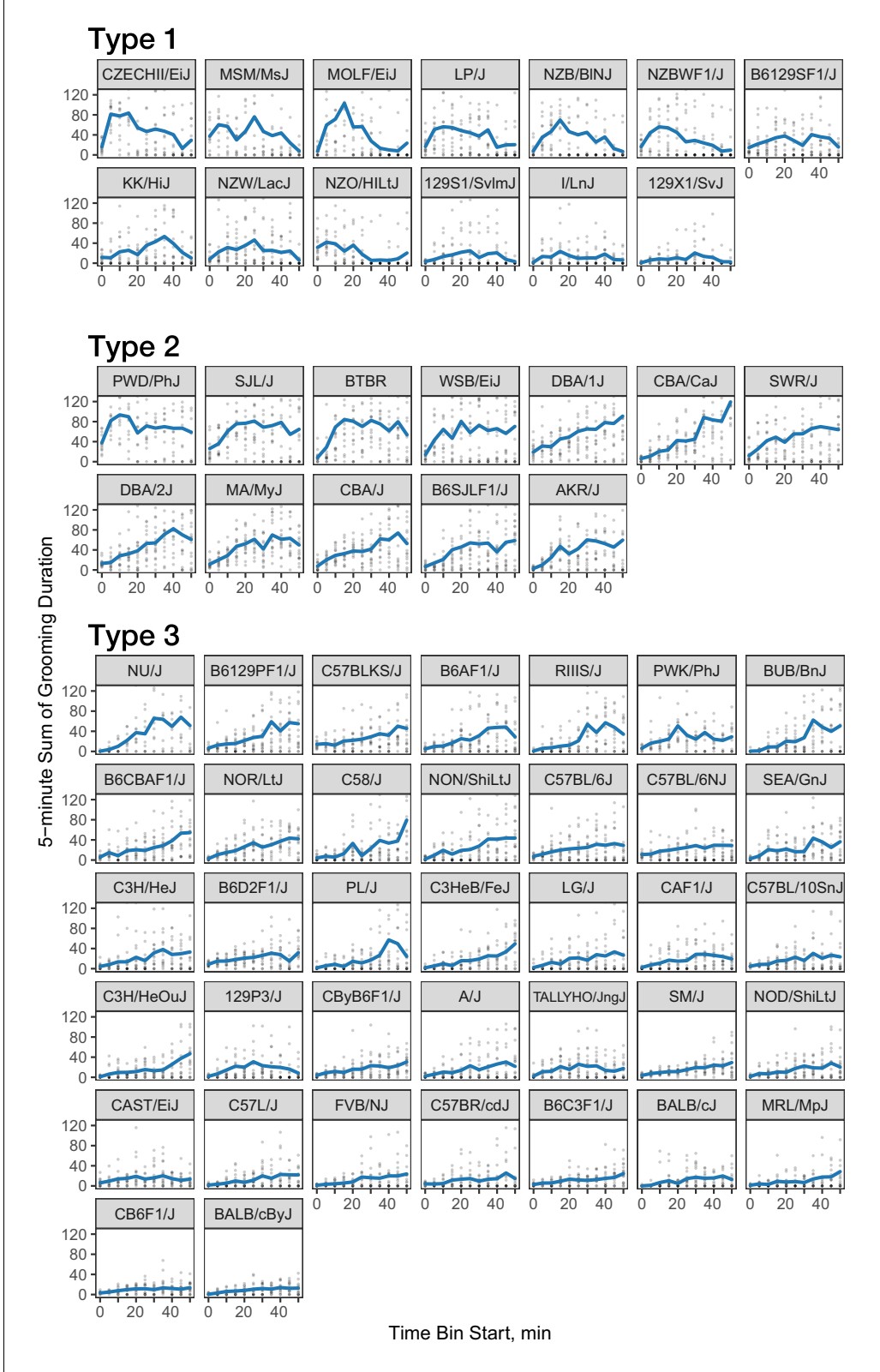

**Figure 8.** Three classes of grooming patterns in the open field revealed by clustering. Grooming duration in 5 min bins is shown over the course of the open field experiment (blue line) and data from individual mice (gray points).

The online version of this article includes the following figure supplement(s) for figure 8:

*Figure 8 continued on next page*

*Figure 8 continued*

**Figure supplement 1.** K-means clustering of grooming patterns Results from the k-means clustering.

promising candidates for further research (*Newman and Girvan, 2004*). Modularity scores of the modules ranged from 0.028 (module 8) to 0.103 (module 1). Second, we used Simes' test to combine the p values of genes to obtain an overall p value for the association with each Psychiatric trait. Then the median of association (-log10(Simes p value)) was calculated in each detected module for prioritization. Using this method, module one again ranked at the top of eight modules (median = 5.29) (*Figure 10—figure supplement 1*). Module one is primarily composed of temperament and personality phenotypes, including neuroticism, mood swings, and irritability traits. Genes in this module have a high level of pleiotropy in both human PheWAS and mouse GWAS. Eight of the 10 most pleiotropic genes from the PheWAS analysis are in this module. Genes in this module include SOX5 noted above, RANGAP1 with 31 associations, and EP300 with 23 significant human phenotypic associations. In conclusion, PheWAS analysis links conserved genes that regulate mouse grooming and other open field behaviors to human psychiatric phenotypes. These human phenotypes include personality traits, addiction, and schizophrenia. This analysis links mouse and human traits through shared underlying genetics and allows us to prioritize gene modules for future work, while serving as a framework for future analysis.

## Discussion

Grooming is an ethologically conserved, neurobiologically important behavior that is often used as an endophenotype for psychiatric illnesses. It is a prototypical stereotyped, patterned behavior with highly variable posture and temporal length. Tools to automatically quantify behaviors such as grooming are needed by the behavioral research community and can be leveraged to gain insight into other complex behaviors (*Spruijt et al., 1992*; *Kalueff et al., 2010*). We present a neural network approach toward automated vertebrate model organism behavioral classification and ethogram generation that achieves human-level performance. This approach is simple, scalable, and can be carried out using standard open field apparatus, and is expected to be of use to the behavioral neuroscience community. Using this grooming behavior classifier, we analyzed a large data set consisting over 2200 hr of video from dozens of mouse strains. We demonstrated environmental effects on grooming, patterns of grooming behavior in the laboratory mouse, carry out a mouse GWAS, and a human PheWAS to understand the underlying human-relevant genetic architecture of grooming and open field behavior in the laboratory mouse.

While the machine learning community has implemented a wide variety of solutions for human action detection, few applications have been applied to animal behavior. This may be in part due to the wide availability of human action data sets and the stringent performance requirements for human bio-behavioral research. We observed that the cost of achieving this stringent performance is very high, requiring a large quantity of annotations. More often than not, experimental paradigms are limited by cost to be short or small enough to cost less to allow manual annotation of the data.

Machine learning approaches have been previously applied to automated annotation of behavioral data. We observed that our 3D convolutional neural network outperforms a JAABA classifier when trained on the same training data set. Our neural network achieved 91.9% true positive rate with a 5% false positive rate while thee JAABA classifier achieved 64.2% true positive rate at a 5% false positive rate. This improvement makes the neural network solution usable for application to biological problems. This improvement is not uniform over all samples but is instead localized to certain types of grooming bouts. This suggests that although the JAABA classifier is powerful, it may be most useful for smaller and more uniform data sets. Experimental paradigms and behaviors with diverse expression will require a more powerful machine learning approach.

With the grooming classifier, we determined the genetic and environmental factors that regulate this behavior. In a large data set collected over an 18-month period using two reference strains, C57BL/6J and C67BL/6N, we assessed effects on grooming of several fixed and dynamic factors including, sex, strain, age, time of day, season, tester, room origin, white noise, and body weight. All mice were housed in identical conditions for at least a week prior to testing. As expected, we

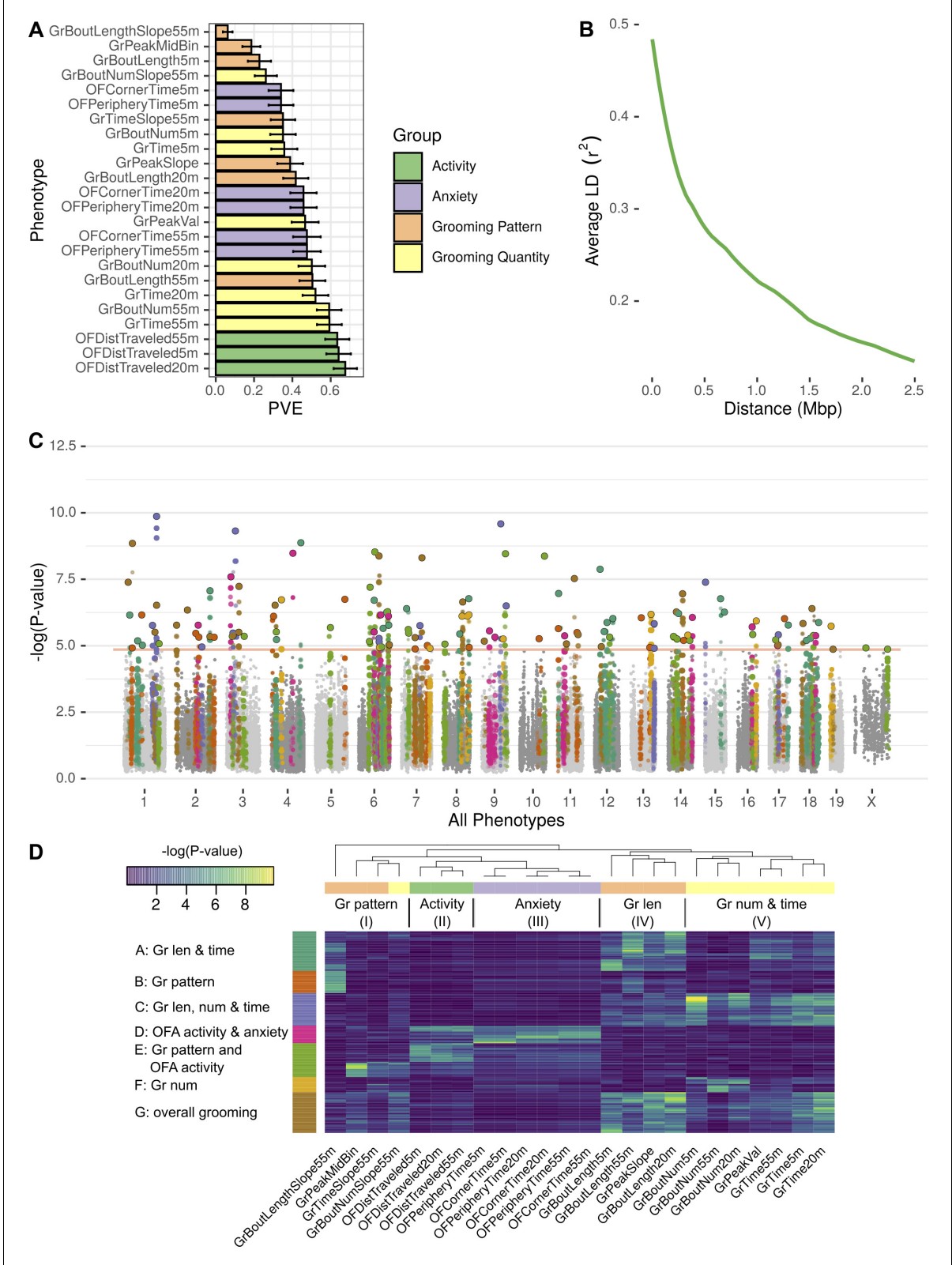

**Figure 9.** GWAS analysis of grooming and open field behaviors. (**A**) Heritability (PVE) estimates of the computed phenotypes. (**B**) Linkage disequilibrium (LD) blocks size, average genotype correlations for SNPs in different genomic distances. (**C**) GWAS results shown as a Manhattan plot of all of the phenotypes combined, colors are according to peak SNPs clusters (from D), all the SNPs in the same LD block are colored according to the

*Figure 9 continued on next page*

Figure 9 continued

peak SNP. Minimal p-value over all of the phenotypes for each SNP (D) Heatmap of all the significant peak SNPs for each. Each row (SNP) is colored according to the assigned cluster in the k-means clustering. The color from k-means cluster is used in C.

The online version of this article includes the following figure supplement(s) for figure 9:

**Figure supplement 1.** Manhattan plot for individual phenotypes.
**Figure supplement 2.** Manhattan plot for individual phenotypes.
**Figure supplement 3.** Manhattan plot for individual phenotypes.
**Figure supplement 4.** Manhattan plot for individual phenotypes.
**Figure supplement 5.** Manhattan plot for individual phenotypes.
**Figure supplement 6.** Manhattan plot for individual phenotypes.

observe strong effects of sex, time of day, and season. In many but not all studies, and not in the present study, tester effects have been observed in the open field in both mice and rats (*Walsh and Cummins, 1976*; *McCall et al., 1969*; *Bohlen et al., 2014*; *Lewejohann et al., 2006*). A recent study demonstrated that male experimenters or even clothes of males elicit stress responses from mice leading to increased thigmotaxis (*Sorge et al., 2014*).

To our surprise, we found that room of origin had a strong effect on grooming behavior of C57BL/6J and C57BL/6N. We did not see a clear directionality of effect between shipping mice and those bred in an adjacent room, and in some cases the effect size was high (z > 1). We hypothesize that this may be due to room-specific stress which has previously been demonstrated to alter grooming (*Kalueff et al., 2016*). Presumably, all external mice had similar experience of shipping from the production rooms to the testing area where they were housed identically for at least 1 week prior to testing. Thus, the potential differential stress experience was in the room of origin where the mouse was born and held until shipping. It is important to note that this shipping was only across buildings on the same campus, and shipping that involves air freight may have more drastic effects. This is a point of caution for use of grooming behavior as an endophenotype. Although we acclimated shipped mice to at least 1 week in an adjacent holding room prior to testing, a longer acclimation period may be required prior to testing.

We carried out a large strain survey to characterize and account for genetic diversity in grooming behavior in the laboratory mouse. We found three types of grooming patterns under our test conditions. Type 1 consists of mice that escalate and deescalate grooming within the 55 min open field test. Strains in this group are often sleeping by the end of the assay, indicating a low arousal state toward the end of the assay. We hypothesize that these strains use grooming as a form of successful de-arousal, a behavior that has been previously noted in rats, birds, and apes (*Spruijt et al., 1992*; *Delius, 1970*). Similar to Type 1, Type 2 groomers escalate grooming quickly to reach peak grooming; however, this group does not deescalate grooming during our assay. We hypothesize that these strains need a longer time or may have a deficiency in de-arousal under our test conditions. Type 3 strains escalate for the duration of the assay indicating they have not reached peak grooming under our assay conditions. BTBR is a member of the Type 2 group with prolonged high levels of grooming from an early point, perhaps indicating a hyperaroused state, or an inability to de-arouse. BTBR mice have previously been shown to have high arousal states and altered dopamine function which may lead to the sustained high levels of grooming (*Squillace et al., 2014*). We postulate that other strains in the Type 2 grooming class may also show phenotypic features of ASD, warranting further study of ASD-related phenotypes in these strains.

Wild derived strains have distinct patterns of grooming compared to classical strains. Wild-derived strains groom significantly more and have longer grooming bouts. In our grooming clustering analysis, most of the wild derived strains belong to Type 1 or 2, where as most classical strains belong in Type 3. In addition to *M.m domesticus*, the wild derived inbred lines we tested represent *M.m musculus, M.m castaneous, and M.m molossinus* subspecies. Even though there are dozens of classical inbred strains, there are approximately 5 million SNPs between any two classical inbred laboratory strains such as C57BL/6J and DBA2J (*Keane et al., 2011*). Indeed, over 97% of the genome of classical strains can be explained by fewer than 10 haplotypes indicating small number of classes within which all strains are identical by descent with respect to a common ancestor (*Yang et al., 2011*). Classical laboratory strains are derived from mouse fanciers in China, Japan and Europe before being co-opted for biomedical research (*Morse, 1978*; *Silver, 1995*). Wild derived inbred

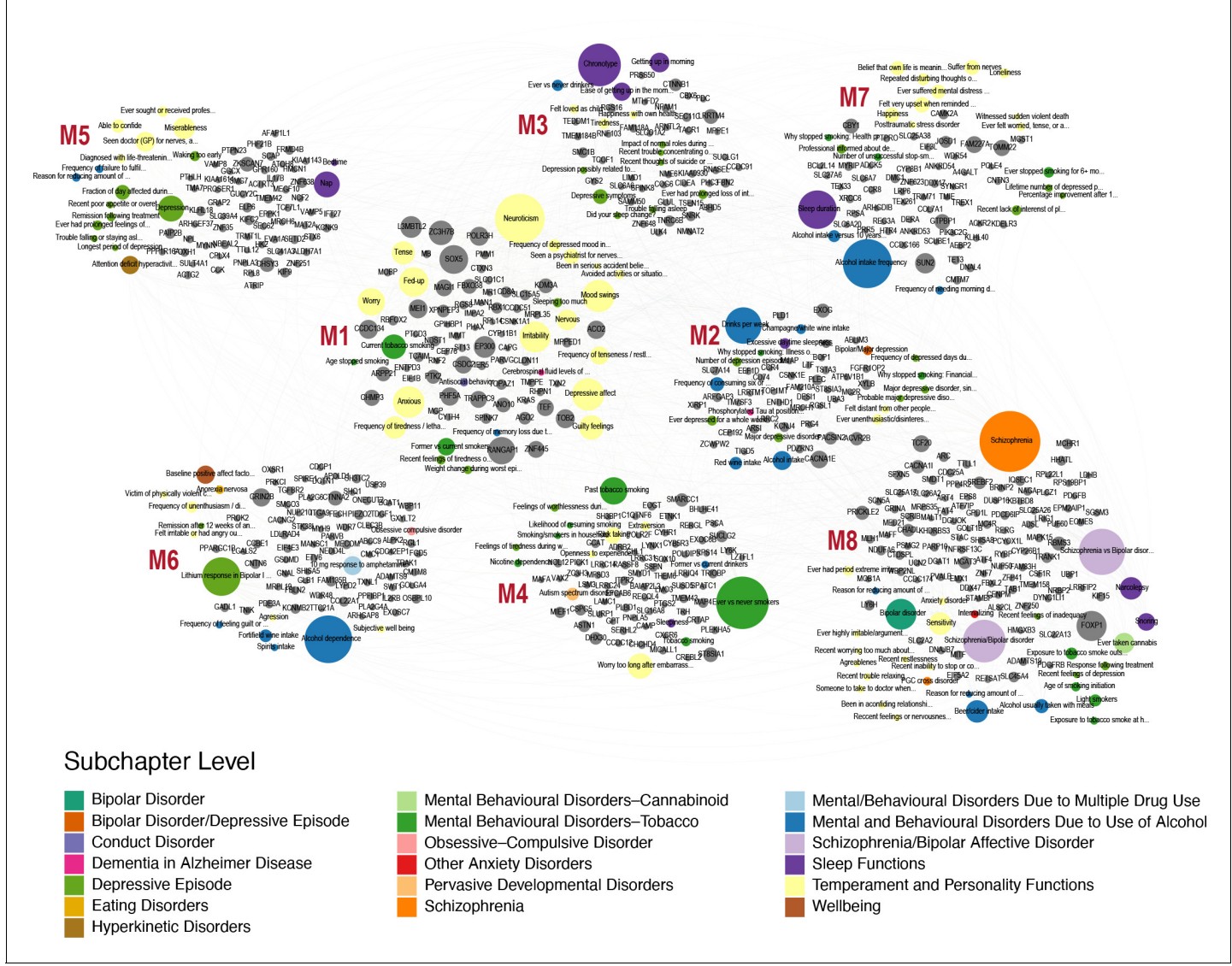

**Figure 10.** Human-Mouse trait relations through weighted bipartite network of PheWAS results. We identified 509 human orthologs out of 860 mouse genes and focused on their association with traits in the Psychiatric domain of gwasATLAS with gene-level p value ≤ 0.001. We represented the relationships between these genes and Psychiatric traits by a weighted bipartite network. The width of an edge between a gene node (gray color) and a Psychiatric trait node is proportional to the association strength (-log10(p value)). The size of a node is proportional to the number of associated genes or traits and the color of a trait node corresponds to the subchapter level in the Psychiatric domain. Eight modules were identified and visualized using Gephi 0.9.2 software.

The online version of this article includes the following figure supplement(s) for figure 10:

**Figure supplement 1.** Box plot of Simes P values in eight modules.

strains such as CAST/EiJ and PWK/PhJ have over 17 million SNPs compared to B6J. Thus, the seven wild derived strains we tested represent far more of the genetic diversity present in the natural mouse population than the numerous classical inbred laboratory strains. Behaviors seen in the wild-derived strains are more likely to represent behaviors in the natural mouse population.

Mouse fanciers breed mice for visual and behavioral distinctiveness, and many exhibit them in competitive shows. Mouse fanciers judge mice on 'condition and temperament' and suggest that 'it is useless to show a mouse rough in coat or in anything but the mouse perfect condition' (*Davies, 1912*). Much like dogs and horses, the 'best individuals should be mated together regard-less of relationship as long as mice are large, hardy, and free from disease' (*Davies, 1912*). It is

plausible that normal levels of grooming behavior seen in wild mice was considered unhygienic or indicative of parasites such as lice, ticks, fleas, or mites. High grooming could be interpreted as poor condition and would lead the mouse fancier to select mouse strains with low grooming behaviors. This selection could account for low grooming seen in the classical laboratory strains.

We used the strain survey data to conduct a mouse GWAS which identified 130 QTL that regulates heritable variation in open field and grooming behaviors. We found that the majority of the grooming traits are moderately to highly heritable. A previous study using the BXD recombinant inbred panel identified one significant locus on chromosome four that regulates grooming and open field activity (*Delprato et al., 2017*). We closely analyzed 862 genes in the QTL interval that are highly pleiotropic and find enriched pathways that regulate neuronal development and function. We then associated those intervals to one of seven clusters which regulate combinations of open field and grooming phenotypes. Mouse grooming can be used as a model of human grooming disorders such as tricotillomania; however, grooming is regulated by the basal ganglia and other brain regions and can be used more broadly as an endophenotype for many psychiatric traits, including ASD, schizophrenia, and Parkinson's (*Kalueff et al., 2016*). We conducted a PheWAS with the highly pleiotropic genes and identified human psychiatric traits that are associated with these genes. This approach allowed us to link mouse and human phenotypes through the underlying genetic architecture. This approach linked human temperament, personality traits, schizophrenia, and bipolar disorder to mouse phenotypes. Future research is needed to definitively link mouse genetic variants to altered behavior. Our GWAS results are a starting point for understanding the genetic architecture of grooming behavior and will require functional studies in the future to assign causation.

In conclusion, we describe a neural network based machine learning approach for action detection in mice and apply it towards grooming behavior. Using this tool, we characterized grooming behavior and its underlying genetic architecture in the laboratory mouse. Our approach to grooming is simple and can be carried out using standard open field apparatus and should be of use to the behavioral neuroscience community.

## Materials and methods

### Animals

All animals were obtained from The Jackson Laboratory production colonies or bred in a room adjacent to the testing room as previously described (*Geuther et al., 2019*; *Kumar et al., 2011*). Partial data of the strain survey was published previously and reanalyzed for grooming behavior (*Geuther et al., 2019*). All behavioral tests were performed in accordance with approved protocols from The Jackson Laboratory Institutional Animal Care and Use Committee guidelines.

### Data set annotation

We selected data to annotate by training a preliminary JAABA classifier for grooming, then clipping video chunks based on predictions for a wide variety of videos. The initial JAABA classifier was trained on 13 short clips that were manually enriched for grooming activity. This classifier is intentionally weak, designed simply to prioritize video clips that would be beneficial to select for annotation. We then applied this weak classifier on a larger library of videos. The video clips are a subset of our previously published dataset and include 157 individual mouse videos that represent 60 different mouse strains (*Geuther et al., 2019*). We clipped video time segments with 150 frames surrounding grooming activity prediction to mitigate chances of a highly imbalanced data set. We generated 1253 video clips which total 2,637,363 frames. Each video had variable duration, depending upon the grooming prediction length. The shortest video clip contains 500 frames, while the longest video clip contains 23,922 frames. The median video clip length is 1348 frames. Also see *Supplementary file 1* for additional annotated dataset metadata.

From here, we trained seven annotators. From this pool of seven trained annotators, we assigned two annotators to annotate each video clip completely. If there was confusion for a specific frame or sequence of frames, we allowed the annotators to request additional opinions. Annotators were required to provide a 'Grooming' or 'Not Grooming' annotation for each frame, with the intent that difficult frames to annotate would get different annotations from each annotator. We only train and validate using frames in which annotators agree, which reduces the total frames to 2,487,883.

## Neural network model

Our neural network follows a typical feature encoder structure except using 3D convolution and pooling layers instead of 2D. We started with a $16 \times 112 \times 112 \times 1$ input video frames, where 16 refers to the time dimension of the input and one refers to the color depth (monochrome). Each convolution layer that we applied is zero-padded to maintain the same height and width dimension. Additionally, each convolution layer is followed by batch normalization and ReLU activation. First, we applied two sequential 3D convolution layers with a kernel size of $3 \times 3 \times 3$ and number of filters of 4. Second, we applied a max pooling layer of shape $2 \times 2 \times 2$ to result in a new tensor shape of $8 \times 64 \times 64 \times 4$. We repeated this two 3D convolution and max pool, doubling the filter depth each time, an additional three more times which results in a $1 \times 8 \times 8 \times 32$ tensor shape. We applied two final 3D convolutions with a $1 \times 3 \times 3$ kernel size and 64 filter depth, resulting in a $1 \times 8 \times 8 \times 64$ tensor shape. Here, we flattened the network to produce a $64 \times 64$ tensor. After flattening we applied two fully connected layers, each with 128 filter depth, batch normalization, and ReLU activations. Finally, we added one more fully connected layer with only two filter depth and a softmax activation. This final layer was used as the output probabilities for not grooming and grooming predictions for the last of the 16 frames.

## Neural network training

We trained four individual neural networks using the same training set and four independent initializations. During training, we randomly sample video chunks from the data set where the final frame contains an annotation where the annotators agree. Since we sample a 16 frame duration, this refers to the 16th frame's annotation. If a frame selected does not have 15 frames of video earlier, the tensor is padded with 0-initialized frames. We apply random rotations and reflections of the data, achieving an 8x increase in effective data set size. The loss function we use in our network is a categorical cross entropy loss, comparing the softmax prediction from the network to a one-hot vector with the correct classification. We use the Adam optimizer with an initial learning rate of $10^{-5}$. We apply a decay schedule of learning rate to halve the learning rate if five epochs persist without validation accuracy increase. We also employ a stop criteria if validation accuracy does not improve by 1% after 10 epochs. During training, we assemble a batch size of 128 example video clips. Typical training would be done after 13–15 epochs, running for 23–25 epochs without additional improvement.

## JAABA training

We trained JAABA classifiers using two different approaches. Our first approach was using the guidelines provided by the software developers (JAABA Interactive Training). This involves interactively and iteratively training classifiers. The data selection approach is to annotate some data, then prioritize new annotations where the algorithm is unsure or incorrectly making predictions. We continued this interactive training until the algorithm no longer made improvements in a k-fold cross validation.

Our second approach was to subset our large annotated data set to fit into JAABA and train on the agreeing annotations. Initially, we attempted to utilize the entire training data set, but our machine did not have enough RAM to handle the entire training data set. The workstation we used contained 96 GB of available RAM. We wrote a custom script to convert our annotation format to populate annotations in a JAABA classifier file. To confirm our data was input correctly, we examined the annotations from within the JAABA interface. Once we created this file, we could simply train the JAABA classification using JAABA's interface. After training, we applied the model to the validation data set to compare with our neural network models. We repeated this with various sizes of training data sets.

## Definition of grooming behavioral metrics

Here, we describe a variety of grooming behavioral metrics that we use in following analyses. Following the approach that has previously been proposed, we define a single grooming bout as a duration of continuous time spent grooming without interruption that exceeds 3 s (*Kalueff et al., 2010*). We allow brief pauses (less than 10 s), but do not allow any locomotor activity for this merging of time segments spent grooming. Specifically, a pause occurs when motion of the mouse does

not exceed twice its average body length. In order to reduce the complexity of the data, we summarize the grooming duration, number of bouts, and average bout duration into 1 min segments. In order to have a whole number of bouts per time duration, we assign grooming bouts to the time segment when a bout begins. In rare instances where multiple-minute bouts occur, this allows for a 1 min time segment to contain more than 1 min worth of grooming duration.

From here, we sum the total duration of grooming calls in all grooming bouts to calculate the total duration of grooming. Note that this excludes un-joined grooming segments less than 3 s duration as they are not considered a bout. Additionally, we count the total number of bouts.

Once we have the number of bouts and total duration, we calculate the average bout duration by dividing the two. Finally, we bin the data into one minute time segments and fit a linear line to the data. Positive slopes for total grooming duration infer that the individual mouse is increasing its time spent grooming the longer it remains in the open field test. Negative slopes for total grooming duration infer that the mouse spends more time grooming at the start of the open field test than at the end. This is typically due to the mouse choosing to spend more time doing another activity over grooming, such as sleeping. Positive slopes for number of bouts infer that the mouse is initiating more grooming bouts the longer it remains in the open field test.

For k-means clustering of the behavior across strains, we visually inspected the data and decided three clusters was optimal. We z-score transformed grooming features as inputs to the k-means algorithm to determine cluster membership. Finally, we projected the clusters discovered by the k-means to a 2D space formed by principal components (PC).

## Data collection and reporting

Protocols for data collection were previously described in *Geuther et al., 2019*. In brief, each animal was video recorded from a top-down viewpoint for 55 min of novel open field exposure. Imaging parameters were held constant across videos, including camera distance, zoom, frame rate, and background conditions. No power analysis was used sample size for C57BL/6NJ vs C57BL/6J data since this data is longitudinal control data. Power analysis for the strain survey data showed that with 16 animals (8M/8F), we have 80% power to detect a effect size of 1 (Cohen's d). Outliers in the strain survey were removed when individuals measured a value of $GrTime55m < Q_1 - 1.5 * IQR$ or $GrTime55m > Q_3 + 1.5 * IQR$, where $Q_1$ is the first quartile, $Q_3$ is the third quartile, and $IQR$ is the interquartile range.

All behavioral data will be available in the Mouse Phenome Database (MPD), and code and models will be available in Kumar Lab Github account (https://github.com/KumarLabJax and https://www.kumarlab.org/data/).

## Genome wide association study (GWAS)

The phenotypes obtained by the machine learning algorithm for several strains were used to study the association between the genome and the strains behavior. A subset of ten individuals from each combination of strain and sex were randomly selected from the tested mice to ensure equal within group sample sizes. The genotypes of the different strains were obtained from the mouse phenome database (https://phenome.jax.org/genotypes). The Mouse Diversity Array (MDA) genotypes were used, di-allele genomes were deduced from parent genomes. SNPs with at least 10% MAF and at most 5% missing data were used, resulting with 222,967 SNPs out of 470,818 SNPs genotyped in the MDA array. LMM method from the GEMMA software package (*Zhou and Stephens, 2012*) was used for GWAS of each phenotype with the Wald test for computing the p-values. A Leave One Chromosome Out (LOCO) approach was used, each chromosome was tested using a kinship matrix computed using the other chromosomes to avoid proximal contamination. Initial results showed a wide peak in chromosome seven around the Tyr gene, a well-known coat-color locus, across most phenotypes. To control for this phenomenon, the genotype at SNP rs32105080 was used as a covariate when running GEMMA. Sex was also used as a covariate. To evaluate SNP heritability, GEMMA was used without the LOCO approach. The kinship matrix was evaluated using all the SNPs in the genome and GEMMA LMM output of the proportion of variance in phenotypes explained – the PVE and the PVESE were used as chip heritability and its standard error.

To estimate the LD decay, pairs of SNPs that are at most 2.5 Mbp apart had their genotypes correlation computed using Pearson correlation. The pairs of SNPs were divided into bins according to

their distance, each bin size being 5000 bp. The average correlation $r^2$ coefficient of pairs of SNPs in each bin were averaged and plotted against the average SNPs distance and smoothed using loess function. Correlation of $r^2 > 0.2$ was set as a threshold for assigning SNPs to the same QTL. QTL were determined by sorting the SNPs according to their p-values, then, for each SNP, determining a locus centered at this SNP by adding other SNPs with high correlation ($r^2 > 0.2$) to the peak SNP. Each locus was limited to 10 Mbp from the initial peak SNP selected upstream and downstream. The peak SNPs were aggregated from all the phenotypes and the p-values were used to cluster the peaks into clusters using the k-means algorithm implemented in R. We repeated k-means clustering for a variety of number of clusters and visually decided that seven clusters was appropriate for this data. The GWAS results of phenotypes in each cluster were combined by taking the minimal p-value of all the phenotypes in each cluster, for each SNP. The entire set of phenotypes was also combined in the same manner.

The GWAS execution was wrapped in an R package called mouseGWAS available on github: https://github.com/TheJacksonLaboratory/mousegwas; *Geuther, 2021*; copy archived at swh:1:rev:5d2caac2637da442f4b9648ac1eb1f35bd1136cf it also includes a singularity container definition file and a nextflow pipeline for regenerating the results. The command used for generating the results is:

```
export  GH=https://raw.githubusercontent.com/TheJacksonLaboratory/mousegwas/
grooming
nextflow run TheJacksonLaboratory/mousegwas -r grooming \
-yaml $GH/example/grooming_nowild.yaml \
-shufyaml $GH/example/grooming_shuffle.yaml \
-input $GH/example/grooming_paper_strain_survey_2019_11_21.csv \
-outdir grooming_output –addpostp="-loddrop 0" -profile slurm,singularity.
```

## Human phenome-wide association study (PheWAS)

In order to carry out such cross-species association and link the mouse genetic circuit of grooming to human phenotypes, we conducted a phenome-wide association study (PheWAS). First, we identified human orthologs of the 860 mouse grooming and open field genes with at least 6 degrees of pleiotropy. For each human ortholog, we downloaded PheWAS summary statistics from gwasATLAS (https://atlas.ctglab.nl/, release 2: v20190117) (*Watanabe et al., 2019*). We focused on the association in the Psychiatric domain with gene-level p value $\leq 0.001$. Second, in order to visualize and cluster these associations, we represented the relationships between genes and psychiatric traits by a weighted bipartite network, in which the width of an edge between a gene node and a Psychiatric trait node is proportional to the association strength (-log10(p value)). The size of a node is proportional to the number of associated genes or traits and the color of a trait node corresponds to the subchapter level in the Psychiatric domain. To identify modules within this network, we applied an improved community detection algorithm for maximizing weighted modularity in weighted bipartite networks (*Dormann and Strauss, 2014*), using bipartite R package (*Dormann et al., 2008*). All the networks were visualized using Gephi 0.9.2 software (*Bastian et al., 2009*).

## Acknowledgements

We thank Drs. Kristin Branson and Mayank Kabra (Janelia Research Campus) for providing initial directions on the project and help with JAABA. We thank Drs. Greg Carter and Daniel Skelly for critical feedback on the manuscript. We thank members of the Kumar Lab for helpful advice and Taneli Helenius for editing. We thank JAX Information Technology team members Edwardo Zaborowski, Shane Sanders, Rich Brey, David McKenzie, and Jason Macklin for infrastructure support. This work was funded by The Jackson Laboratory Directors Innovation Fund, National Institute of Health DA041668 (NIDA), DA048634 (NIDA), and Brain and Behavioral Foundation Young Investigator Award (VK). This work used the National Science Foundation (NSF) Extreme Science and Engineering Discovery Environment (XSEDE) XStream service at Stanford University through allocation TG-DBS170004 (to VK).

## Additional information

### Funding

| Funder | Grant reference number | Author |
| --- | --- | --- |
| Jackson Laboratory | Director's Innovation Fund | Vivek Kumar |
| National Institutes of Health | DA041668 | Vivek Kumar |
| National Institutes of Health | DA048634 | Vivek Kumar |
| National Science Foundation | TG-DBS170004 | Vivek Kumar |
| Brain and Behavior Research Foundation | | Vivek Kumar |

The funders had no role in study design, data collection and interpretation, or the decision to submit the work for publication.

### Author contributions

Brian Q Geuther, Conceptualization, Data curation, Software, Formal analysis, Validation, Investigation, Visualization, Methodology, Writing - original draft, Writing - review and editing; Asaf Peer, Software, Formal analysis, Validation, Visualization, Methodology, Writing - original draft; Hao He, Formal analysis, Investigation, Visualization; Gautam Sabnis, Formal analysis, Investigation; Vivek M Philip, Formal analysis, Supervision, Methodology, Writing - original draft; Vivek Kumar, Conceptualization, Supervision, Funding acquisition, Methodology, Writing - original draft, Project administration, Writing - review and editing

### Author ORCIDs

Brian Q Geuther (iD) https://orcid.org/0000-0002-7822-486X
Asaf Peer (iD) https://orcid.org/0000-0002-7577-353X
Vivek Kumar (iD) https://orcid.org/0000-0001-6643-7465

### Ethics

Animal experimentation: All studies were performed in accordance with approved protocols from The Jackson Laboratory Institutional Animal Care and Use Committee guidelines (Animal Protocol Number 14010).

### Decision letter and Author response

Decision letter https://doi.org/10.7554/eLife.63207.sa1
Author response https://doi.org/10.7554/eLife.63207.sa2

## Additional files

### Supplementary files

• Supplementary file 1. Full metadata related to the created annotated dataset. Metadata columns includes strain, sex, arena, testing date, weight, training/validation split, animal number, starting frame for clip, duration of clip, clip identity in released dataset, and clip identity referenced in this paper.

• Supplementary file 2. Full results of the GWAS and PheWAS.

• Transparent reporting form

### Data availability

The all machine learning datasets are available here: https://www.kumarlab.org/2021/03/11/grooming-behavioral-data/. The code is available here: https://github.com/KumarLabJax/MouseGrooming (copy archived at https://archive.softwareheritage.org/swh:1:rev:

f18b268d7a4aba4cf7f5aad72ae5438dfdb311cf/). Behavioral data has been deposited into Mouse Phenome Database. The access for this data will be https://mpdpreview.jax.org/projects/Project1051.

The following datasets were generated:

| Author(s) | Year | Dataset title | Dataset URL | Database and Identifier |
|---|---|---|---|---|
| Geuther BQ | 2021 | Mouse Grooming Detection Annotated Dataset | https://zenodo.org/record/4646088 | Zenodo, 10.5281/zenodo.4646088 |
| Geuther BQ | 2021 | Grooming Behavioral Strain Survey Data | https://phenome.jax.org/projects/Kumar3 | Mouse Phenome Database, Kumar3 |

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
