## [Decision Letter]

**Acceptance summary:**

We congratulate you on your development of a neural network that automatically identifies grooming in mice. Your work to characterize grooming behavior across strains, sex and conditions for C57 mice (e.g. lighting, season), to identify loci and genes linked to grooming and open field behavior and to identify human homologs for genes linked to grooming in mice will have sustained influence on the field.

**Decision letter after peer review:**

Thank you for submitting your article "Action detection using a neural network elucidates the genetics of mouse grooming behavior" for consideration by *eLife*. Your article has been reviewed by two peer reviewers, and the evaluation has been overseen by a Reviewing Editor and Michael Taffe as the Senior Editor. The following individuals involved in review of your submission have agreed to reveal their identity: Ioana Carcea (Reviewer #2); Nancy Padilla (Reviewer #3).

The reviewers have discussed the reviews with one another and the Reviewing Editor has drafted this decision to help you prepare a revised submission.

Summary:

In summary, Geuther et al. demonstrate the use of machine learning to identify and classify grooming behaviors in an impressively varied cohort of mice species. Using a 3D convolutional neural network that automatically identifies grooming in mice, they conducted a genome-wide association study (GWAS) to identify loci and genes linked to grooming and open-field behavior. Finally, the authors performed analyses to identify human homologs for genes linked to grooming in mice, with a focus on loci associated with psychiatric illness in humans. Novel findings include the identification of gene-phenotype modules that identify genes linked to both human and mouse phenotypes.

Essential revisions:

1) It is unclear how robust the grooming algorithm is to new videos/new animals. Were the training video clips all from a small subset of videos? The authors say 2M frames were annotated from 1,253 video segments, but it is unclear how many strains/mice/videos (not video clips or segments) they came from in total. In the Materials and methods, they mention that the training data comes from 1,253 video clips but there are no additional details. How many animals were represented in the training dataset and in the testing dataset? How robust is the algorithm to differences in animal size (which can be affected by camera distance) and to video frame rate? Where all videos used with this network taken at identical camera distances, video frame rates, and backgrounds (e.g. home cage vs. white background)? Since the training dataset needed for good performance is so large (2M frames) understanding the flexibility of the network is crucial for the community to adopt it successfully. When using and implementing the network, the devil is on the details.

2) The authors train JAABA with 20% of their dataset and show in Figure 3A and supplementary figure that it performs worse than the new network with the same 20%, but AUCs are relatively similar. But to be sure that this is not a fluke of the specific subset of the data used and to have statistical power to claim the superiority the authors should subsample the training data and repeat the comparison more times. This way they can determine if there the network with 20% of the data outperforms JAABA in every case or with X probability.

3) The authors use k-means clustering for different analyses, but there are no details on how the different clustering procedures were done.

4) Reviewers have concerns related to the robustness of the 3D convolutional neural network. We believe that in order for the scientific community to take advantage of this tool the authors need to provide more information.

---

## [Author Response]

Essential revisions:1) It is unclear how robust the grooming algorithm is to new videos/new animals. Were the training video clips all from a small subset of videos? The authors say 2M frames were annotated from 1,253 video segments, but it is unclear how many strains/mice/videos (not video clips or segments) they came from in total. In the Materials and methods, they mention that the training data comes from 1,253 video clips but there are no additional details. How many animals were represented in the training dataset and in the testing dataset? How robust is the algorithm to differences in animal size (which can be affected by camera distance) and to video frame rate? Where all videos used with this network taken at identical camera distances, video frame rates, and backgrounds (e.g. home cage vs. white background)? Since the training dataset needed for good performance is so large (2M frames) understanding the flexibility of the network is crucial for the community to adopt it successfully. When using and implementing the network, the devil is on the details.

We thank the reviewers for pointing these out and we agree that it is important to provide detailed information about the training data. The annotated video data were sampled from our published strain survey dataset (Geuther, 2019). All videos in this dataset were imaged using nearly identical hardware and as such frame rate, camera distance, and a white open field arena background are held constant. We expect that it is necessary to utilize techniques such as transfer learning to allow this network to generalize well outside these environmental conditions.

As per the reviewers’ request, we have expanded the details on the training dataset. In the new supplementary figure for Figure 2 (Figure 2—figure supplement 1), we provide visual descriptions for training dataset metadata. Additionally, we now include a table containing metadata for the entire annotated dataset (TrainingDatasetMeta_SupTable.zip). The 1253 video clips were sampled from 157 videos each which contained a different mouse. These 157 mice represent 60 different strains and a wide range of body weights (11g – 44g). While all validation clips also had training clips sampled from the same video, they were non-overlapping and separated in time.

We have added details of the video data collection and new supplementary data to the text of the paper.

2) The authors train JAABA with 20% of their dataset and show in Figure 3A and supplementary figure that it performs worse than the new network with the same 20%, but AUCs are relatively similar. But to be sure that this is not a fluke of the specific subset of the data used and to have statistical power to claim the superiority the authors should subsample the training data and repeat the comparison more times. This way they can determine if there the network with 20% of the data outperforms JAABA in every case or with X probability.

We have added an additional supplementary figure panel to address this concern. Figure 3—figure supplement 1 now contains a second panel to show how the performance of JAABA scales with different training dataset sizes. Additionally, we also now report AUC values for all ROC plots. We observe that performance saturates for JAABA when using 10% of our annotated data, by only achieving an increase of 0.005 AUC (from 0.925 to 0.93) while doubling the size of training data. Interestingly, JAABA using 5% outperforms our neural network using 10%. This data shows that although JAABA may perform better using limited small datasets, both a neural network approach and a larger training dataset is necessary for generalizing on larger and more varied data.

Related edits are located in the subsection “Proposed Neural Network Solution” and the new panel B for Figure 3—figure supplement 1 in the figure legend. We have also added indicator to Figure 3B, to indicate the True Positive at 5% False Positive for JAABA and Neural Network solution. The improvement seen in the neural network solution is large (62% to 92% TP at 5% FP, Figure 3B pink line).

3) The authors use k-means clustering for different analyses, but there are no details on how the different clustering procedures were done.

We have added text in the Materials and methods to describe methods surrounding Figure 8’s clustering. Briefly, we visually inspected the data and decided there should be 3 clusters. Then, we used zscore transformed features as inputs to the k-means algorithm to determine cluster membership. We projected the clusters discovered by the k-means to a 2D space formed by principal components (PC).

We have adjusted the Materials and methods text of Figure 9D to better describe the clustering of grouped SNPs. Briefly, we ran the analysis for a variety of clusters and visually selected the results which looked to cluster the best visually.

4) Reviewers have concerns related to the robustness of the 3D convolutional neural network. We believe that in order for the scientific community to take advantage of this tool the authors need to provide more information.

We interpret “robustness” and “need to provide more information” at two levels. As in critique 1, robustness implies how well does the network work on visual diversity (small and large animals, light vs. dark animals). The network we train is one of the most robust in the field since it recognizes behavior across high visual diversity of the mouse. The network we have trained provides accurate classification across a diversity of mouse visual appearance (coat color, size etc). In fact we are not aware of another work that attempts to operate across all diverse mouse strains for action detection. Furthermore, the network can always be improved for new strains or environments by adding examples to the training data.

The comment could also refer to whether the architecture of the network has been optimized. This could be done by adding more layers or modifying the structure of the architecture. If we added more layers or made the network larger or smaller would we still obtain the same level of accuracy? We already achieve human observer level performance and don’t expect to exceed that performance, a limitation set by the human generated training data. We believe that identifying a more efficient network is important, but also outside the scope of this paper. The field of machine learning network optimization which seeks to maintain accuracy while minimizing compute is a subfield of machine learning and will require further research. These are important future work.